# The first map of crop sequences types in Europe over 2012-2018

Rémy Ballot[1], Nicolas Guilpart[1], Marie-Hélène Jeuffroy[1]

[1] Université Paris-Saclay, AgroParisTech, INRAE, UMR Agronomie, 78850, Thiverval-Grignon, France

*Correspondence to*: Rémy Ballot (remy.ballot@inrae.fr)

**Abstract.** Crop diversification is considered as a key element of the agroecological transition, whereas current dominant cropping systems are known to rely only on a few crops species – like cereals in Europe. To assess the benefits of crop diversification at large scale, an accurate description of current crop sequences is required as a baseline. However, such a description is lacking at the scale of Europe. Here, we developed the first map of dominant crop sequences in Europe for the period 2012-2018. We used the LUCAS dataset that provides temporally-incomplete (2012, 2015 and 2018) land cover information from a stable grid of points covering Europe. Eight crop sequence types were identified using hierarchical clustering implemented on LUCAS data, and mapped over Europe. We show, in France, that the relative importance of these eight crop sequence types (as estimated from LUCAS data) was highly consistent with those derived from an almost spatially-exhaustive temporally-complete national dataset (the French Land Parcel Identification System) for the same period, thus validating the method and the typology for this country. Land use (i.e. crop production area) derived from our map of dominant crop sequences was also highly consistent with land use reported by official statistics, both at national and Europe levels, validating the approach at Europe-scale. This first map of dominant crop sequences in Europe should be useful for future studies dealing with agricultural issues that are sensitive to crop rotations. The map of dominant crop sequences types in Europe derived from our work is available at https://doi.org/10.5281/zenodo.7016986 (Ballot et al., 2022).

## 1 Introduction

Crop diversification – the increase of crop diversity from field to national scale – is considered as a key element of the agroecological transition (Kremen et al., 2012; Lechenet et al., 2016; Renard and Tilman, 2021; Wezel et al., 2014), whereas current crop sequences are short and specialized in many parts of the world (Meynard et al., 2018; Salembier et al., 2016; Schott et al., 2010; Stein and Steinmann, 2018). Indeed, crop diversification supports a number of ecosystem services, with positive effects on crop yield, soil fertility, nutrient cycling, carbon sequestration, climate and water regulation, pest control, biodiversity, and pollination (Beillouin et al., 2019, 2021a; Tamburini et al., 2020). Strategies of crop diversification practices include crop rotation, cultivar mixture, cover crops, intercropping and agroforestry. Among these strategies, diversifying the crop sequence is of particular interest in order to: (i) increase nutrient availability and limit fertilizer requirement (e.g. by the inclusion of legume crops), (ii) favor soil protection and conservation by enhancing soil cover, (iii) promote natural regulation of pests and diseases by avoiding the presence of successive host crops, and (iv) reduce weed infestation (Wezel et al., 2014).

The term crop sequence refers to the sequence of crops grown in succession in the same field over a given period of time (Dury et al. 2012). A crop sequence is then defined by the nature of its crops and their order of succession. Based on this definition, the temporal frequencies of crops is a key feature of a crop sequence. The term crop rotation is also commonly used to refer to the sequence of crops grown in succession in the same field (Bullock, 1992), but includes a notion of cyclicality (e.g. rotation length) – at least to some degree (Castellazzi et al. 2007). Hereafter, we use the term crop sequence rather than crop rotation

because we consider a fixed period of time (i.e. 2012-2018), focus on temporal frequencies of crops, and do not analyze cyclicality in crop sequences.

Aiming at leveraging benefits from diversifying crop sequences, several foresight studies have assessed scenarios for the future of agriculture in which crop sequences are modified to increase the services they provide (Billen et al., 2021; Poux and Aubert, 2018), or scenarios of organic farming expansion where significant differences in crop rotations between conventional and

organic agriculture are acknowledged (Barbieri et al., 2017, 2019, 2021). A robust assessment of benefits expected from crop diversification at large scale requires an accurate description of current crop sequences as a baseline. Indeed, several ecosystem services provided by diversified farming systems depend on the pre-crop effect, thus needing to know precisely the nature of the crops successively grown on a same field (Bennett et al., 2012). However, such a description is still lacking for Europe, for at least two reasons: (i) existing datasets at the European scale, e.g. the LUCAS dataset, provide information about land

use categories and crop species cultivated on agricultural land, but no information about crop sequences because data are not available every year (e.g. the LUCAS dataset provides information every 3 years), (ii) even if national and sub-national datasets describing crop sequences are available within some EU-state members, e.g. the Land Parcel Identification Systems in France (Levavasseur et al., 2016), a lack of harmonization (e.g. spatial and temporal resolution or nomenclature) between these datasets makes them difficult to use in a cross-analysis. This situation has resulted in the production of regional to national

studies of crop sequences (Levavasseur et al., 2016; Peltonen-Sainio and Jauhiainen, 2019; Stein and Steinmann, 2018; Xiao et al., 2014) but has hampered any analysis at the scale of the European Union (EU). Such an analysis would be useful, especially in the context of the EU's Farm to Fork strategy, in which the adoption of more diverse crop rotations is encouraged (European Commission, 2022). To overcome this problem, we developed an original method that combines European-level and national-level datasets to create a map of current dominant crop sequences at the European level. We show that the

temporally-incomplete (i.e. every 3 years) information on crop sequences provided by the LUCAS dataset can be used to derive robust estimates of crops frequencies in the sequence when compared to crop frequencies derived from a temporally-complete national-level dataset. We acknowledge this approach does not allow capturing the exact order of crops in the succession. Nonetheless, it allows capturing crop frequencies which are a key feature of crop sequences (Castellazzi et al. 2007; Peltonen-Sainio and Jauhiainen 2019).

## 2. Materials and Methods

### 2.1 Method overview

The purpose of this work is to map current dominant crop sequences from the European Land Use Cover Area frame statistical Survey (LUCAS). For this study, the multi-year harmonised data by d'Andrimont et al. (2020) was used. As this dataset is temporally incomplete (observations of land use on fixed points, only every three years), we proceeded in three steps (Figure 1) to assess how this incomplete information could be used to describe the diversity and localization of major crop sequences across Europe. First, we filtered the LUCAS data, by selecting points under non-perennial agricultural land cover. For each of these points, we calculated frequencies of eight crops or groups of crops across 2012, 2015 and 2018 (the three last years of observation). For example, a point identified as wheat in 2012 and corn in 2015 and 2018 was converted into: cereal frequency = 0,33, corn frequency = 0,67. Second, based on this set of eight variables, LUCAS points were classified into eight groups (hereafter referred to as crop sequence types), combining a principle component analysis (PCA) with hierarchical clustering. Third, we assessed the consistency of this classification against crop sequences derived with the quasi-exhaustive yearly French Land Parcel Identification System (LPIS), and with the Eurostat crop production data. All data sources used in this study are detailed in Table 1.

### 2.2. Data sources

### 2.2.1. The LUCAS dataset

The Land Use Cover Area frame Survey (LUCAS) is an in-situ land-cover (i.e. physical cover observed at the earth's surface, e.g. cereals, root crops, fodder crops) and land-use (i.e. socio-economic function of the observed earth's surface, e.g. agriculture, industry, residential) observation, carried out on approximately 300 000 points sampled from a stable grid of around 1 100 000 points across Europe since 2006, every three years. Last observation campaign was realized in 2018. The work presented here is based on the harmonized LUCAS database developed by d'Andrimont et al. (2020), which gathered, in a unique dataset, all the information collected from the beginning of this survey (Table 1). All EU-28 member states are represented in this dataset, except Croatia and Malta. Montenegro, which is not a member state, is also represented, as well as Great Britain, which is no longer a member state. Thus, "Europe" will be use hereafter to refer to the spatial scope of this study.

### 2.2.2. The French LPIS dataset

The French Land Parcel Identification System (LPIS) is based on yearly declarations made by farmers in compliance with Common Agricultural Policy subsidies. From 2008, it provides an annual almost spatially-exhaustive information of land use for agricultural land, detailing 28 crop categories until 2014, and approximately 300 from 2015. Until 2014, information was collected at the block scale. Each block can enclose one or more agricultural parcels, and thus one or more crops with declared area for each one but no geolocalisation within the block. Due to parcels reconfiguration from one year to another, it is not

straightforward to know pluriannual crop sequences from the French LPIS, for years older than 2015. For our study, we used the "RPG Explorer Crop successions France version 2.0" dataset developed by Martin et al. (2021, Table 1). This dataset compiles all annual LPIS data for France into a unique dataset of crop sequences for more than 18 million fields over the period 2012-2018, and associated areas. It also relies on an algorithm, which identifies the most-likely crop sequence when more than one crop is declared for one given block. This dataset represents the most exhaustive dataset of crop sequences in France, in terms of both spatial and temporal resolution.

### 2.2.3. The Eurostat dataset

The Eurostat crop production dataset provides harmonized information about annual crop-specific acreage per country from the year 2000 (EUROSTAT, 2023, Table 1). The crop statistics are collected by the National Statistical Institutes and/or Ministries of Agriculture by using several statistical methods: sample surveys, administrative sources, expert estimates. Most often a combination of several methods is used. Eurostat is independent from LUCAS data.

### 2.3 Identification and mapping of dominant crop sequence types in the EU based on LUCAS data

### 2.3.1 Preprocessing of LUCAS and French LPIS data

From the harmonized LUCAS database (d'Andrimont et al., 2020), we selected points with observations for the three most recent campaigns at time of the study (i.e. 2012, 2015, and 2018). We assumed the seven-year time period long enough to encompass the duration of main crop rotations, and thus we do not consider older campaigns (i.e. 2006 and 2009) which may be outdated to represent current crop sequences. Another reason why we considered only the three most recent campaigns, is that if the whole 2006-2018 period was considered, only 9 094 points would have remained with information about crop cultivated in 2006, 2009, 2012, 2015 and 2018. This choice would also have limited the analysis to 11 among the 27 EU countries (i.e. Belgium, Czechia, Germany, Spain, France, Hungary, Italy, Luxembourg, The Netherlands, Poland and Slovakia).

We discarded points labelled as non-agricultural use or permanent agricultural use (e.g. orchards, vineyards) in at least one year among 2012, 2015 and 2018 (i.e. Land Cover (LC) not included in land cover classification B11 to B55). As an exception, we also conserved 2 609 points identified as permanent grasslands in 2018 (i.e. LC E20), but identified as a non-permanent agricultural use in 2012 or 2015, as they may be the result of a confusion between temporary and permanent grassland during observation. This resulted in a dataset of 21 620 points, with information about crops cultivated in 2012, 2015 and 2018. Thus, each point is associated with a single (temporally incomplete) crop sequence.

Depending on the purpose of analysis, crop sequences can be described in many different ways, including length and flexibility of rotation (Castellazzi et al., 2008), nature and function of crops or crop groups (Barbieri et al., 2017; Videla-Mensegue et al., 2022), diversity of crops or crop groups (Beillouin et al., 2021b; Tamburini et al., 2020), diversity of sowing dates (Weisberger et al., 2019), order of succession of crops within the sequence (Peltonen-Sainio et al., 2019), and temporal frequency or return

time of crops within the sequence (Nowak et al., 2022). Given the characteristics of the LUCAS dataset, the exact order of crop succession within the sequences could not be described, nor could be the variety of sowing dates (e.g. the LUCAS dataset provides no information about whether the crop is sown in autumn or spring). Therefore, we chose to describe current dominant

crop sequences by the temporal frequencies, over three years (i.e. 2012, 2015 and 2018), of eight crops or groups of crops described below. This choice is a compromise between addressing common issues related to crop rotations (e.g. nitrogen management and the role of legumes) and avoiding unnecessary complexity. These eight groups were defined based on two criteria: (i) their agronomic relevance, and (ii) their importance in terms of cultivated area in Europe. Groups considered were (i) cereals (i.e. wheat, barley, oat, triticale, rye, corresponding to LC B11 to B19 except B16 in the LUCAS dataset

nomenclature), (ii) corn (LC B16), (iii) rapeseed (LC B32), (iv) sunflower (LC B31), (v) pulses (i.e. dry pulses and soybean, corresponding to LC B33 and B41), (vi) root crops (i.e. beets and potatoes, corresponding to LC B21 to B23), (vii) forage legumes (i.e. alfalfa and clover, corresponding to LC B51 and B52) and (viii) temporary grassland (LC B53, B55, and E20, Table 2). As these eight groups did not encompass all the land cover categories, the sum of (groups of) crops frequencies may be lower than 1 for each point.

In order to serve the quality assessment step, the French LPIS dataset was preprocessed the same way. First, fields under perennial use were discarded. Then, (groups of) crops frequencies were calculated for each field.

### 2.3.2 Identification of crop sequence types with hierarchical clustering

A principle component analysis (PCA) was performed with the PCA() function of the R package FactoMineR v2.1 (Lê et al., 2008) on all the eight variables describing temporal frequencies of cereals, rapeseed, sunflower, pulses, corn, root crops, forage

legumes, and temporary grassland. Seven components displayed eigenvalues higher than or close to 1, cumulating more than 99% of variance (Figure 2a). These seven components were used as inputs for a hierarchical clustering, performed with the HCPC() function from the same R package. Inertia gain displayed a clear break from eight to nine (Figure 2b), thus eight groups of crop sequences (crop sequence types) were considered. A crop sequence type has thus been assigned to all points considered. To analyze the geographical distribution of dominant crop sequences, observed locations of these eight crop

sequence types were mapped across Europe.

### 2.3.3 Quality assessment

The crop sequence types derived from incomplete temporal sequences (LUCAS dataset) were compared to crop sequence types derived from complete temporal sequences based on the French LPIS dataset. The French LPIS represents the best available data for spatial distribution of crops in France in terms of spatial and temporal resolutions, spatial coverage, and

150 disaggregation by crop type, , with a coverage higher than 98% for all field crops or 92% for temporary grasslands (Cantelaube et Lardot, 2022, Guilpart et al., 2022). To this aim, a three-step procedure was followed. First, a random forest (RF) model was trained on LUCAS data to predict crop sequence type from crop (or crop group) frequencies (i.e. a total of eight predictors). The RF model was fitted using the randomForest() function of the R package randomForest v4.6.14 (Liaw and Wiener, 2002),

with default settings (i.e. 500 trees, two variables randomly sampled as candidates at each split). The RF model showed good performances, as indicated by an out-of-bag (OOB) error rate lower than 0.1% (Table 4). Each tree of our RF model is constructed based on a random sample of the observations generated by bootstrap. The observations that are not part of the bootstrap sample are referred to as OOB observations, which are being used for estimating the prediction error, the so-called OOB error. The OOB error is considered as a good measure of the true prediction error (Matthew, 2011; Janitza and Hornung, 2018). Second, the RF model was applied on French LPIS data to classify observed crop sequences into the eight crop sequence types. Third, the crop sequence type distributions derived from the LUCAS dataset and predicted from French LPIS dataset were compared at the national and regional levels in France. This allowed to check if the lower temporal and spatial resolutions of the LUCAS dataset led to some bias in the identification of crop sequence types and their geographical distribution.

To ensure the map of crop sequence types presented in this paper could be used as a baseline in studies assessing scenarios for the future of agriculture in Europe, it appears essential to assess its consistency with observed land use. To this aim, average crop or crop group harvested areas between 2012 and 2018 were calculated from (i) theirtemporal frequencies within the crop sequence types derived from the LUCAS dataset, (ii) relative importance (i.e. spatial frequency) of crop sequence types, and (iii) total arable land area following Equation (1):

$$harvested\_area_i = \sum_j F_{j,i}. F_j. total\_area \qquad \text{(Eq. 1)}$$

Where $F_j$ is the spatial frequency of crop sequence type $j$, and $F_{i,j}$ is the temporal frequency of (group of) crop $i$ within the crop sequence type $j$. This calculation was applied for each country in Europe, at the national level, and compared to crop- or crop group harvested areas reported by Eurostat (2022). Relative importance of crop sequence types were calculated according to the number of points and without consideration to field area. Indeed, the harmonized LUCAS dataset provides information about size of the surveyed parcels in hectares, but this information is limited to four categories (i.e. < 0,5 ha, 0,5 – 1 ha, 1 – 10 ha, > 10 ha), which were not relevant to weight relative importances.

## 3.    Results

### 3.1    Eight types of crop sequences identified based on the LUCAS dataset

Eight different types of crop sequences were identified based on the LUCAS dataset (Table 3). One crop sequence type largely dominate in terms of relative importance within Europe: it accounts for 35% of LUCAS points, and is dominated by cereals (79%), followed by corn (8%) and grasslands (6%). Then comes a group of four crop sequence types of moderate relative importance, each one representing 10% to 13% of LUCAS points. This group includes a crop sequence type dominated by rapeseed (43%) and cereals (43%), a second one dominated by grasslands (74%), cereals (13%) and corn (11%), a third one dominated by root crops (43%) and cereals (35%) and a fourth one dominated by corn (82%) and cereals (17%). The remaining three crop sequence types can be considered as minor as they individually account for less than 10% of LUCAS points. This includes a crop sequence type dominated by by sunflower (42%) and cereals (39%), a second one dominated by pulses (37%) and cereals (36%), and a third one dominated by forage legumes (47%) and cereals (23%).

Two (groups of) crops are present in all crop sequence types: cereals and corn (Table 2). This highlights the central role of these crops in crop rotations in Europe. On the other hand, pulses, and to a lesser extent sunflower, root crops and forage legumes, appear only in few (i.e. 1 to 3) crop sequence types. Some crop sequence types show a high degree of diversity, like the "pulses and cereals" in which all (groups of) crops are represented, while some others are much less diverse, like the "cereals" and "corn and cereals" types. These two crop sequence types account for almost the half (45%) of all LUCAS points and are composed of cereals and corn by more than 90%.

If all LUCAS campaigns available were considered (i.e. 2006 to 2018), the classification would have resulted in the same eight types of crop sequences. The eight types would have not ranked in the same order, regarding their relative importance. Their relative importance would have changed slightly: 26 instead of 35% for cereals dominated crop sequence type (same ranking), 17% instead of 8% for sunflower and cereals (second instead of sixth place), 16 instead of 10%, for corn and cereals (third instead of fifth place), 13 instead of 11% for temporary grassland dominated crop sequence type (fourth instead of third place), 8 instead of 13% for rapeseed and cereals (fifth instead of second place), 8 instead of 10% for root crops and cereals (sixth instead of fourth place), 7% for pulses and cereals (relative importance and ranking unchanged) and 6 instead of 7% for forage legumes and cereals (ranking unchanged). These differences were consistent as 16 among 27 countries were discarded if we considered only points with observations for all the 2006 to 2018 LUCAS campaigns.

## 3.2    Quality assessment

### 3.2.1    Incomplete temporal crop sequences are a good proxy of complete temporal crop sequences at national and regional scale in France

On average at the national scale in France, the relative importance of crop sequence types estimated from the LUCAS dataset is in good agreement ($R^2$=0.75, RMSE=0.04) with estimates based on the French LPIS dataset, with no systematic bias (Figure 3, Figure S1). However, estimates based on the LUCAS dataset underestimate the proportion of the "grasslands" crop sequence type (-9%), and slightly overestimate the proportion of the "corn and cereals" (+3%), "cereals" (+2,5%), "pulses and cereals" (+2%) and "sunflower and cereals" (-2%) ones.

At the regional scale, the agreement between the relative importance of crop sequence types derived from the LUCAS or French LPIS datasets remains good, but varies by crop group (Figure 4). The crop sequence types "sunflower and cereals", "root crops and cereals", "rapeseed and cereals", "forage legumes and cereals" and "corn and cereals" show the best agreement, with high $R^2$ (higher than 0.8), low RMSE (lower than 0.1), and no systematic bias, whereas the crop sequence type "pulses and cereals" shows the lowest agreement ($R^2$=0.37, RMSE=0.04). Underestimation of the crop sequence type "grasslands" especially for regions where relative importance is high, and overestimation of the crop sequence type "cereals" is confirmed, but for regions where relative importance is low.

Overall, regional differences in crop sequence types are well captured with the LUCAS dataset, as shown by two conclusions. First, any given crop sequence type may have a low relative importance in one region and a high one in another. This range of relative importance across regions is well captured by estimates derived from the LUCAS dataset for all crop sequence types

(Figure 4, Figure S2). Second, regional specificities in terms of dominant crop sequences types are also well captured by LUCAS data (Figure S2). For example, the region Picardie is dominated by three crop sequence types: "root crops and cereals", "cereals", and "rapeseed and cereals"; while the region Aquitaine is dominated by the crop sequence types: "corn and cereals", and "sunflower and cereals".

### 3.2.2 Land use derived from crop sequence types is in good agreement with land use reported by official statistics at national and EU levels

Comparison of crop harvested areas derived either from crop sequence types or from official statistics (EUROSTAT, 2022) shows good agreement at Europe scale (Figure 5), with $R^2$ higher than 0.99 and no bias. Comparison for each country of the EU (Figure 6) reveals good levels of correlation between areas derived from crop sequence types and areas reported by Eurostat with $R^2$ ranging from 0.71 for temporary grasslands, to 0.98 for cereals, rapeseed or pulses. Nevertheless, some crop-country combinations display bad agreement between estimated and observed harvested areas (Figure S3). Harvested area of forage legumes are overestimated in Greece, France or Romania, whereas they are underestimated in Germany or Italy. Regarding corn, harvested area is overestimated for Sweden, Lithuania, Spain or United Kingdom, but underestimated for Hungary and Romania. For root crops, harvested areas is also underestimated for Cyprus, Montenegro, Portugal, Belgium or The Netherlands. Sunflower harvested area is overestimated in Spain and underestimated for Romania. Cereals harvested area is overestimated for The Netherlands, but underestimated for Spain. Temporary grasslands harvested area is overestimated for Romania, Spain, Portugal and Germany.

### 3.3 Spatial distribution of crop sequence types in Europe

The maps of crop sequence types presented in Figure 7 and Figure 8 show a strong spatial pattern of crop sequence types distribution in the EU. The crop sequence types "grasslands", "corn and cereals", "cereals", and, to a lesser extent, "pulses and cereals" and "forage legumes and cereals" are present in all EU countries, whereas the other crop sequence types are concentrated in specific parts of Europe. Indeed, "rapeseed and cereals" and "root crops and cereals" are mainly found in the north of France, Belgium, The Netherlands, Germany, Czech Republic, and Poland, whereas "sunflower and cereals" is mainly found in the south of France, Spain, Hungary, Bulgaria, and Romania. These differences are best highlighted when looking at the latitudinal distribution of crop sequence types shown in Figure 9: Indeed "grasslands", "pulses and cereals", and "cereals" are found in a wide range of latitudes (from 40°N to 60°N) while other crop sequence types are limited to narrower ranges of latitudes with some in the north, like "root crops and cereals" that is concentrated between 50°N and 55°N, and some in the south, like "sunflower and cereals" that is concentrated between 40°N and 50°N. Most crop sequence types display regions in which they are more concentrated, like the north of France and Belgium for "root crops and cereals" or central Italy for "forage legumes and cereals". However, the "pulses and cereals" crop sequence type seems more regularly distributed from 40°N to 60°N. Several crop sequence types coexist in most regions. For example, all crop sequence types except "sunflower and cereals" and "forage legumes and cereals" are found in the very north of France (Figure 8). It is well known that temporal and

spatial crop diversity are not independent from each other (Aramburu Merlos and Hijmans, 2020). Therefore, it is expected that a crop sequence type characterized by a high temporal frequency of a given (group of) crops, will be frequent where this (group of) crops is widely cultivated. For example, crop sequences including corn can only be observed where corn is grown. However, knowing where corn is grown does not tell anything about the crop sequence in which corn is cultivated. Of course,

knowing which other crops are grown in the same area than corn can inform about possible crop sequences, but this is not sufficient. Our results provide a good example: many crop sequence types coexist in the very north of France (temporary grasslands, corn and cereals, root crops and cereals, rapeseed and cereals, pulses and cereals and cereals). As a consequence, cereals (e.g. wheat) can be found in very specialized crop sequences (e.g. the "cereals" crop sequence type), moderately diversified crop sequences (e.g. the "root crops and cereals" crop sequence type) and diversified crop sequences (e.g. the

"pulses and cereals" crop sequence type). This demonstrates that specialized crop sequences can still occur in areas where a substantial diversity of crops is cultivated, and this cannot be inferred from land use (e.g. individual crop maps of "rotation heads") only.

## 4. Discussion

### 4.1 On the quality of the map of crop sequence types

The maps of crop sequence types derived from our study fill an important void in our knowledge of the spatial distribution of crop sequences in Europe. Despite they were based on uncomplete data, both in time and space (the LUCAS dataset), these maps have been shown to represent well the distribution of crop sequence types in France at both national and regional scales when compared to an almost spatially-exhaustive dataset of temporally complete crop sequences. This consistency shows that the temporally incomplete information from LUCAS (i.e. only three crops known – 2012, 2015, and 2018 – from a seven-year

crop sequence) is a good proxy to temporally complete crop sequences at regional to national scales, although individual sequence classification at the field scale may be prone to some bias. At least three different types of bias at the individual field level can be identified.

First, let's consider a three-year cereal-beet-potatoes rotation, which is quite common in North-Western Europe. This rotation may appear in the LUCAS dataset as cereal-cereal-cereal or beet-beet-beet or potato-potato-potato, depending on the crop at

time of observation, and may therefore be classified in the "cereals" or "root crops and cereals" crop sequence type. However, we believe that this individual field classification bias does not affect our results at larger scales as all survey points are not likely to be synchronized in terms of crop sequence.

Second, the comparison with the French LPIS dataset revealed an underestimation of grassland crop sequences, which led to an overestimation of other crop sequence types (Figure 3). A possible explanation for this underestimation could be the inability

of the three sample years available in the LUCAS dataset (i.e. 2012, 2015, 2018) to capture the full diversity of longer crop rotations, such as rotations including temporary grassland. For example, let's consider a cyclical crop rotation starting in 2012 with 3 years of consecutive grassland followed by wheat, wheat, maize and barley. Then, observation in 2012, 2015 and 2018

would be grassland, wheat, and barley respectively, yielding a proportion of grassland of one-third instead of half. Another possible explanation could be that temporary grassland may sometimes be confused with permanent grassland, during LUCAS observations, as well as during farmers' declaration in the French LPIS. In LUCAS dataset, we decided to consider as temporary grasslands points observed as permanent grassland in 2018, but as non-perennial agricultural cover in 2012 and 2015. These points are more likely temporary grasslands confused with permanent ones. But they could also be actual permanent grasslands, after a perennial change in land use between 2015 and 2012. This may contribute to the underestimation of crop sequence with temporary grasslands importance. As highlighted by previous research (e.g. Martin et al. 2020), crop sequences with temporary grasslands provide specific services such as carbon sequestration, biodiversity conservation or weed control. Future users of our dataset should therefore be aware that this underestimation of the importance of crop sequences with temporary grasslands may result in underestimating such services, as well as overestimating the production from other overestimated crops.

Third, the reliability of our validation dataset also needs to be discussed. The French LPIS dataset is based on farmers' declarations, which may not be 100% correct. The RPG explorer algorithm, which compiles LPIS annual raw data into pluriannual crop sequence data, has been validated. However, it cannot be 100% correct, when identifying the most likely crop sequence, when more than one crop was declared per block. A robust estimation of farmers' declaration error and how it could propagate into crop sequence is challenging and could not be done in this study. But to date, the dataset we used is the most complete regarding crop sequences in France.

The comparison between land use derived from our map of crop sequence types and land use derived from official statistics revealed good consistency at Europe level, except for some crop-country combinations. This could be partly explained by the spatial sampling effort of LUCAS data, which can be quantified as the number of LUCAS points per unit area of cropland by country. Analysis of this variable revealed important variation across countries, with the spatial sampling effort ranging from 0.10 (Hungary) to 1.42 (Montenegro) points per thousand hectares of cropland, with a median of 0.22 (Table 5). Accuracy of the map of crop sequence types should therefore be analyzed in the light of this varying spatial sampling effort. For example, our results pointed harvested areas overestimated or underestimated for most (groups of) crops for Romania, which displays a sampling effort lower than the median (0.12). On the other hand, Montenegro displays the highest sampling effort (1.42), but also an important discrepancy between estimated and observed harvested areas. Indeed, this high sampling rate hides a low number of observations (12) for a small arable land. Thus the map accuracy may be limited either by a low sampling effort or by a low number of observations in small-sized countries.

Lastly, we note that the quality assessment of our map of crop sequence types in the EU would benefit from comparisons with other datasets at national or subnational levels in other countries, nevertheless we highlight that all crop sequence types are present in France allowing for a complete quality assessment over the eight crop sequence types.

## 4.2 Perspectives

We assume our maps of crop sequences to be useful for future studies dealing with agricultural issues that are sensitive to crop rotations, including nature-based pest control (Lechenet et al., 2016), pesticide use intensity (Jacquet et al., 2011), nitrogen management and cycling (De Notaris et al., 2018), biodiversity (Li et al., 2021), soil carbon sequestration (King and Blesh, 2018), water resources management (Yang et al. 2015), and crop yield (Bennett et al., 2012). We also highlight that the methodology presented in this paper that leverages temporally incomplete information about crop sequences based on the

LUCAS dataset can be adapted to specific objectives of future studies. For example, looking at the specific place of soybean within crop sequences in the EU would probably require to consider this crop specifically, instead of part of the group "pulses". Different indicators could also be calculated to describe different facets of crop sequences relevant to different ecosystem services. For example, the diversity of crop sowing dates is relevant to weed control (Weisberger et al., 2019), the crop functional diversity is relevant to nitrogen cycling when considering legume and non-legume crops (Zhao et al. 2022), the

phylogenetic crop species diversity might be relevant or not to the control of pests and diseases depending on whether considered crop species are hosts of the same pests and diseases or not (e.g. wheat and barley vs. wheat and soybean) (Beillouin et al. 2021), and the share of summer and winter crops is relevant to water management as water demand is usually higher during the summer. Here we chose the eight crops or groups of crops for their relevance to major agronomical issues (see methods, section 2.3.1). Cereals (wheat, barley, oat, triticale, rye) are all species of the Poaceae family, sharing similar features

regarding their effects on crop sequences. For example, they all have straws with a relatively high C:N ratio, which contributes to limit nitrogen availability for the subsequent crop, when not exported from the field (Justes et al., 2009). Pulses (e.g. pea, fababean, soybean) are all from the Fabaceae family, and have the ability to fix nitrogen from the atmosphere, so that they are usually not fertilized with nitrogen, and contribute to increase nitrogen availability to subsequent crops thanks to the relatively low C:N ratio of their crop residues (Guinet et al., 2020). Root crops (beets and potatoes) may have a specific effect on topsoil

and subsoil structure as belowground organs are harvested (Gabarron-Galeote et al., 2019). Corn, sunflower and rapeseed are three widely grown crops, and have specific characteristics including low pesticide use for sunflower, high insecticide use for rapeseed (Urruty et al., 2016), and high irrigation requirements for maize, especially in the south of Europe (Senthilkumar et al., 2015). Forage legumes and temporary grasslands are commonly cultivated for at least two or three years, and present numerous benefits, including controlling weeds (Martin et al., 2020).

We hope the methodology developed here will stimulate a more detailed description of crop sequences, both in time and space, and their effect on agricultural production and sustainability in future studies.

### Data availability

Data described in this manuscript can be accessed at https://doi.org/10.5281/zenodo.7016986 (Ballot et al., 2022).

**Code availability**

Code used to produce data described in this manuscript, as well as to create figures and tables, can be accessed at https://doi.org/10.5281/zenodo.7018283

**Author contribution**

RB and MHJ initiated research. RB, MHJ and NG designed research. RB performed the analysis with help of NG. RB and NG wrote the manuscript with contributions from MHJ.

**Competing interests**

The authors declare that they have no conflict of interest.

**Acknowledgements**

This work was supported by the LegValue project funded by the European Union's Horizon 2020 research and innovation programme under grant agreement N°727672. Access to some confidential data, on which is based this work, has been made

possible within a secure environment offered by CASD – Centre d'accès sécurisé aux données (Ref. 10.34724/CASD)

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

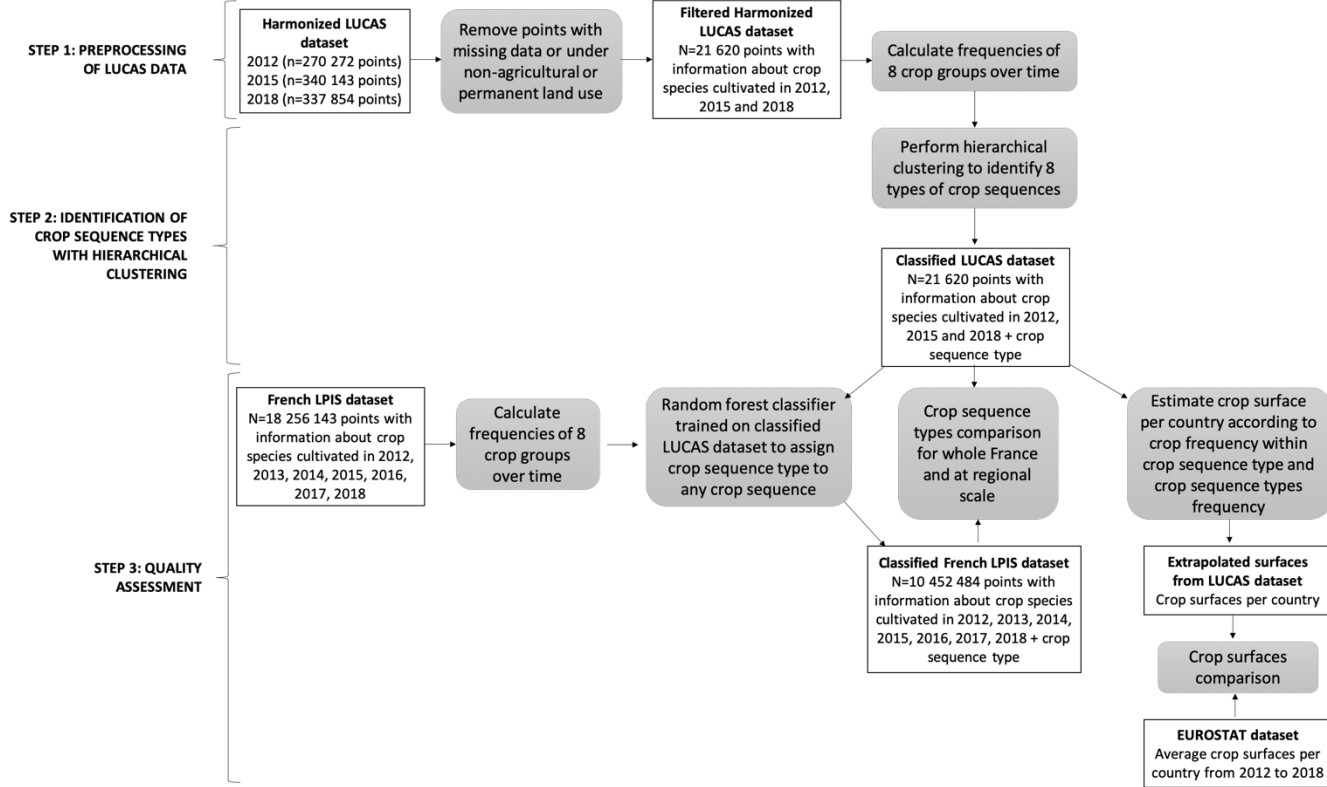

**Figure 1: Workflow diagram for developing spatially-explicit maps of dominant crop sequences across Europe.**

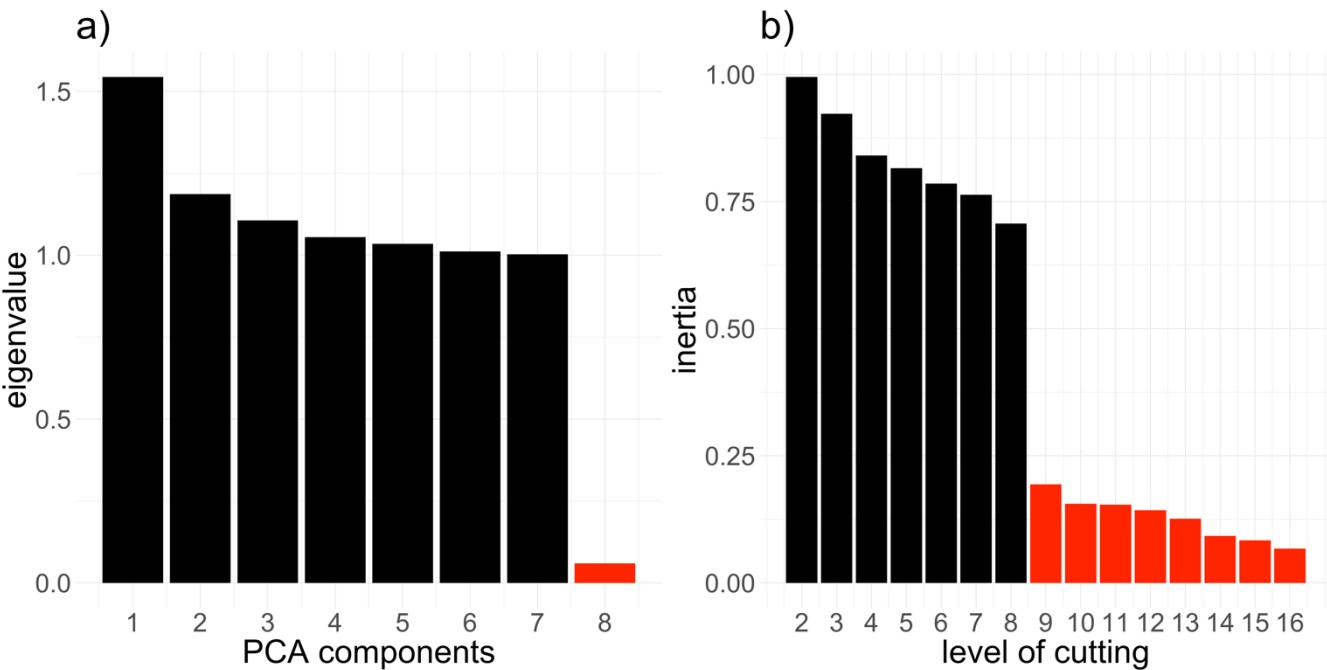

**Figure 2. Identification of crop sequence types based on LUCAS dataset.** (A) Eigenvalues of the components from the Principle Components Analysis. (B) Inertia gain according to the level of cutting. The hierarchical clustering is performed on 43 291 points from the harmonized LUCAS dataset for the years 2012, 2015 and 2018, based on the frequency of 8 crop groups (see text for details).

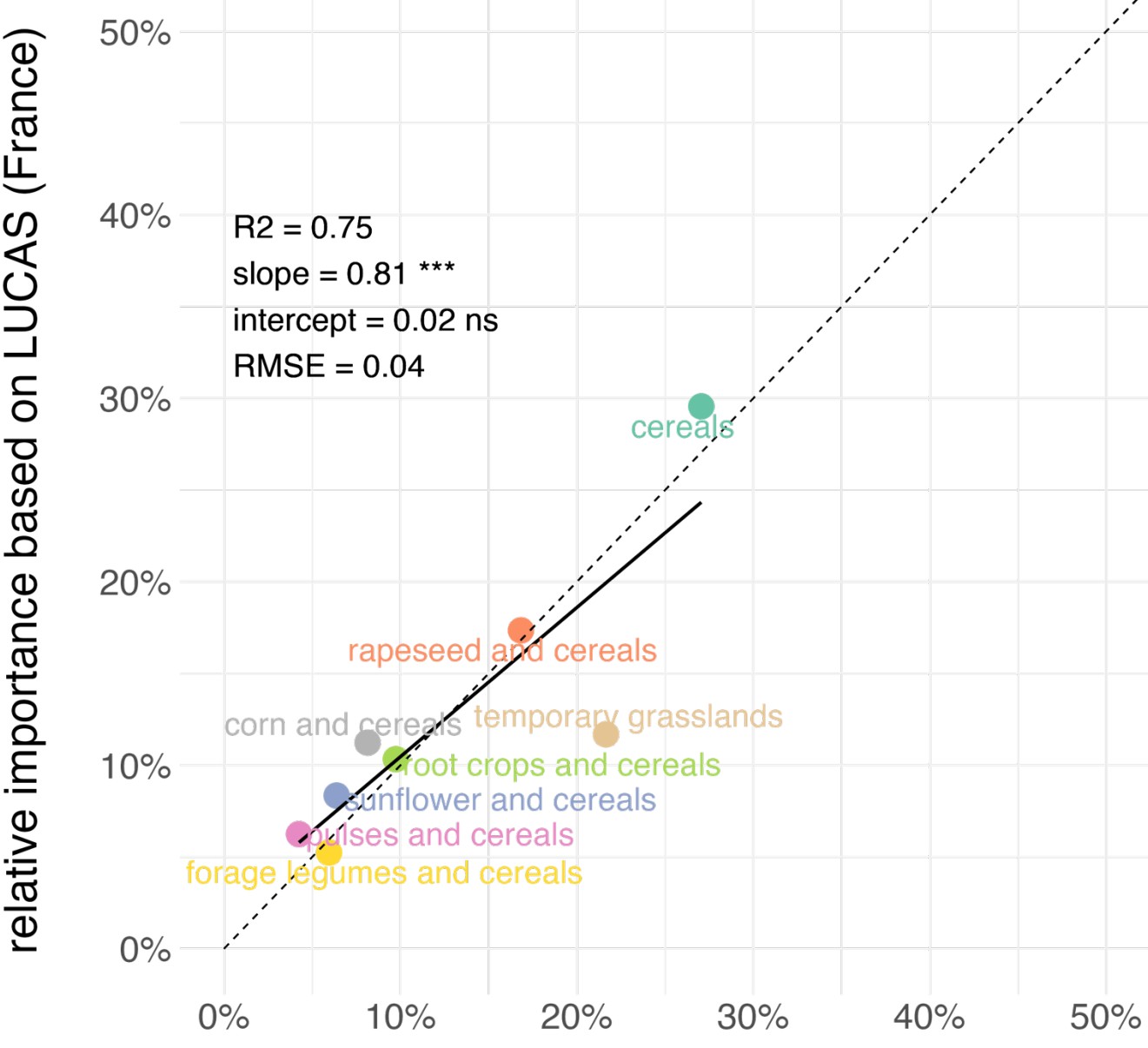

**Figure 3. Comparison of the relative importance of crop sequence types derived from the LUCAS and French LPIS datasets at the national scale in France.** The dotted line denotes the 1:1 line. Also shown are the fitted linear-regression model and the associated slope and intercept. RMSE: Root Mean Square Error.

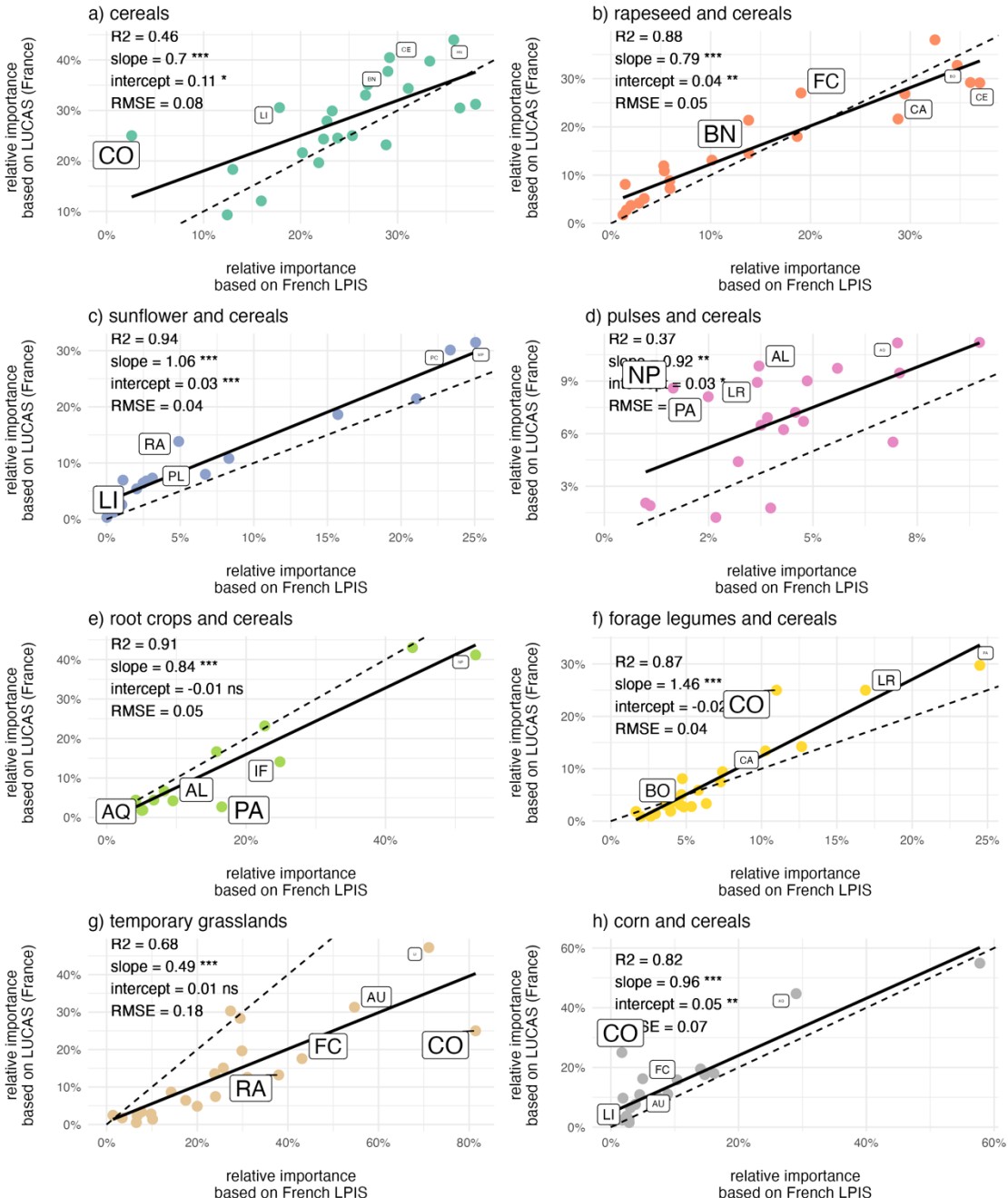

**Figure 4. Comparison of the relative importance of crop sequence types derived from the LUCAS and French LPIS datasets for the 22 regions in France.** The dotted line denotes the 1:1 line. Also shown are the fitted linear regression models and the associated slopes and intercepts. RMSE: Root Mean Square Error. Labels identify the five regions with the highest difference between predicted and observed area (font size is proportional to area difference relative to observed area). AL: Alsace, AQ: Aquitaine, AU: Auvergne, BN: Basse-Normandie, BO: Bourgogne, BR: Bretagne, CE: Centre, CA: Champagne-Ardenne, CO: Corse, FC: Franche-Comté, HN: Haute-Normandie, IF: Île-de-France, LR: Languedoc-Roussillon, LI: Limousin, LO: Lorraine, MP: Midi-Pyrénées, NP: Nord-Pas-de-Calais, PL: Pays de la Loire, PI: Picardie, PC: Poitou-Charentes, PA: Provence-Alpes-Côte d'Azur, RA: Rhône-Alpes

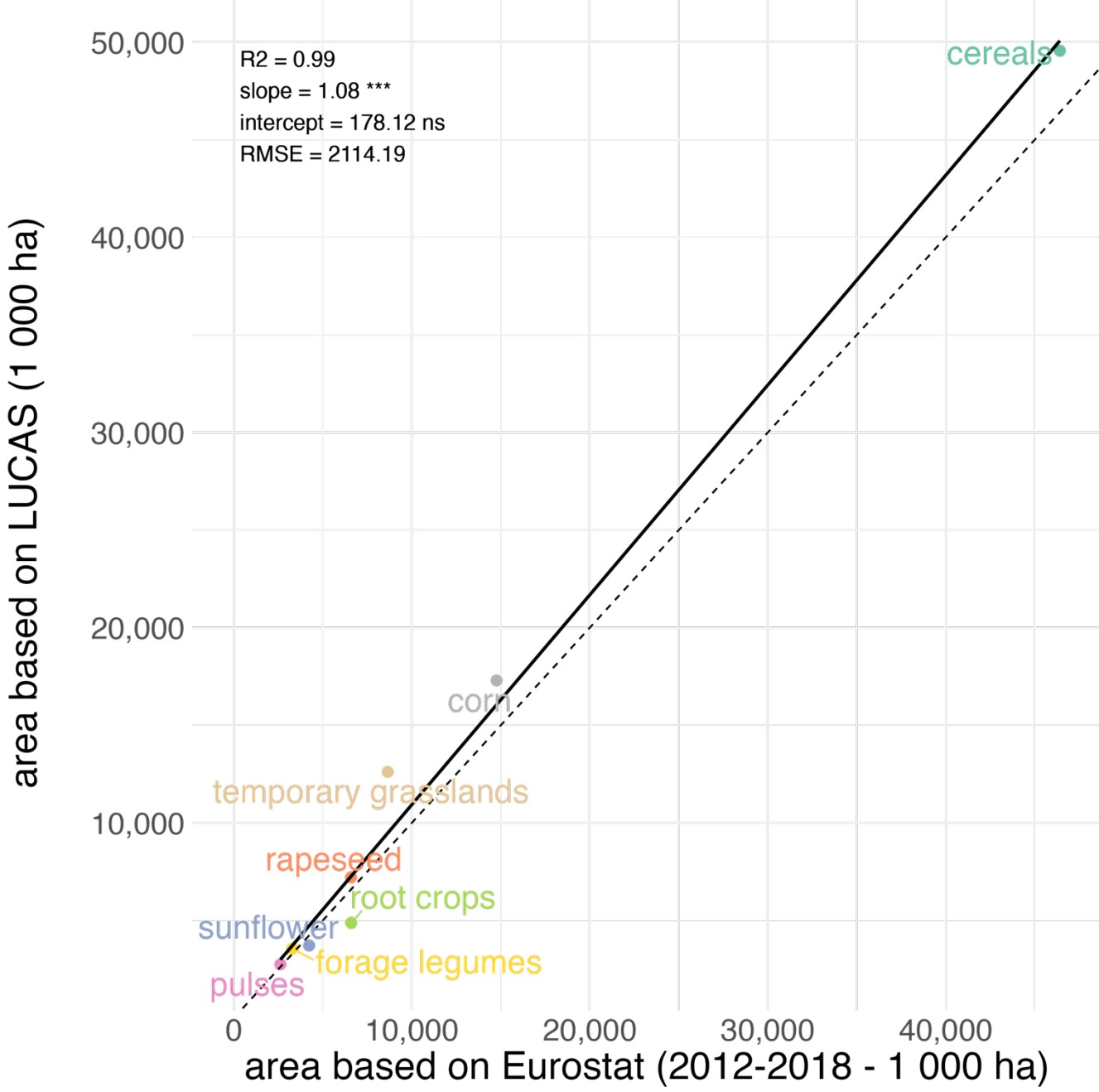

**Figure 5. Comparison of harvested areas reported by Eurostat and derived from crop sequence types based on the LUCAS dataset at Europe scale.** The dotted line denotes the 1:1 line. Also shown is the fitted linear-regression model and the associated slope and intercept. RMSE: Root Mean Square Error.

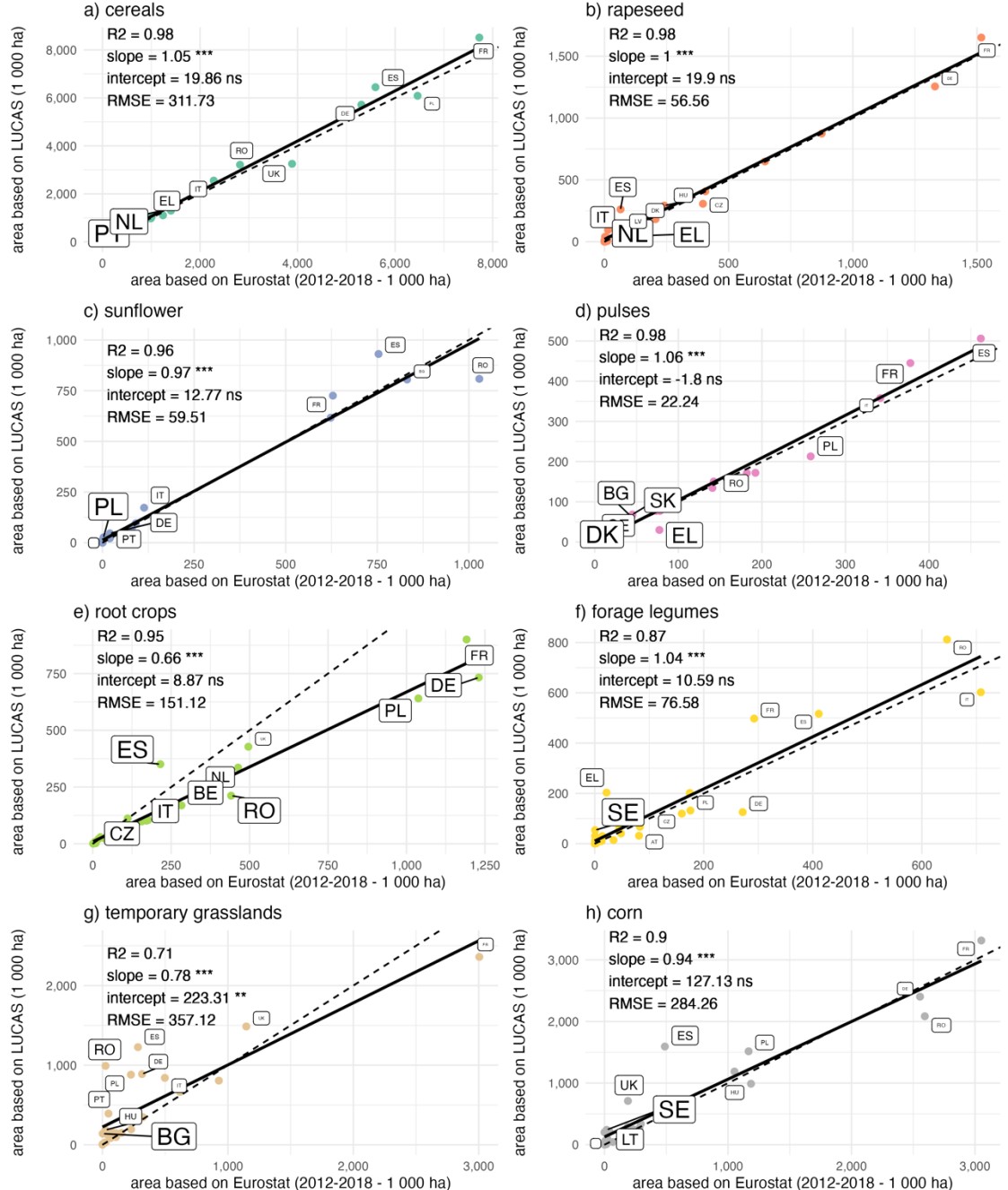

**Figure 6. Comparison of harvested areas reported by Eurostat and derived from the crop sequence types based on the LUCAS dataset, for the various countries in Europe.** The dotted line denotes the 1:1 line. Also shown are the fitted linear-regression models and the associated slopes and intercepts. RMSE: Root Mean Square Error. Labels identify the 10 countries with the highest difference between predicted and observed area (font size is proportional to area difference relative to observed area). AT: Austria, BE: Belgium, BG: Bulgaria, CY: Cyprus, CZ: Czechia, DE: Germany, DK: Denmark, EE: Estonia, EL: Greece, ES: Spain, FI: Finland, FR: France, HU: Hungary, IE: Ireland, IT: Italy, LT: Lithuania, LU: Luxembourg, LV: Latvia, MT: Montenegro, NL: Netherlands, PL: Poland, PT: Portugal, RO: Romania, SE: Sweden, SI: Slovenia, SK: Slovakia, UK: United Kingdom.

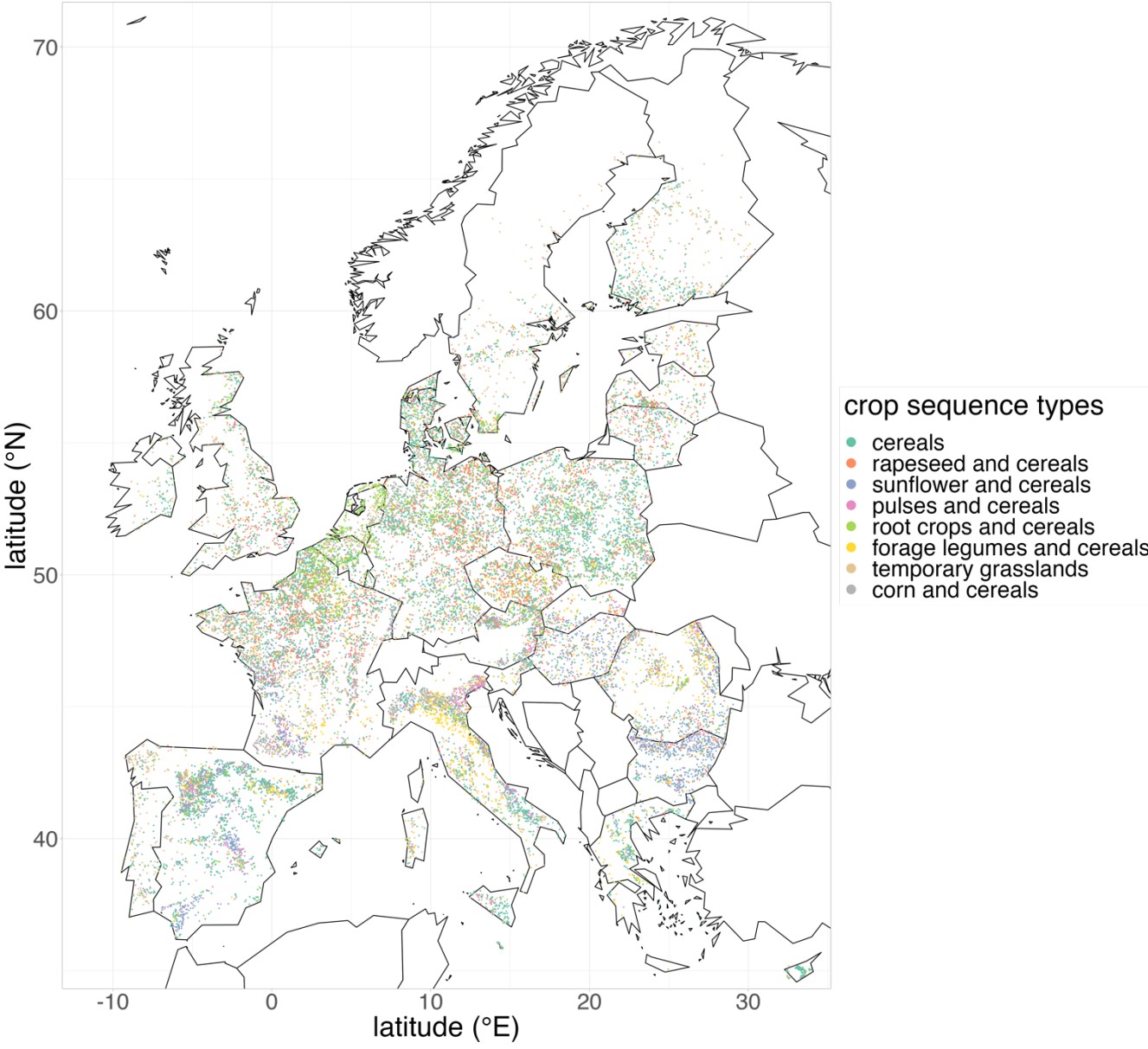

**Figure 7. Map of crop sequence types in Europe based on the LUCAS dataset.** Note that crop sequence types have been given short names that reflect dominant crops or crop groups within the sequence. For example, the crop sequence type "grasslands" also includes cereals and corn, but these crops are less frequent in the sequence. Composition of each crop sequence type in terms of temporal frequencies of (groups of) crops is presented in Table 2.

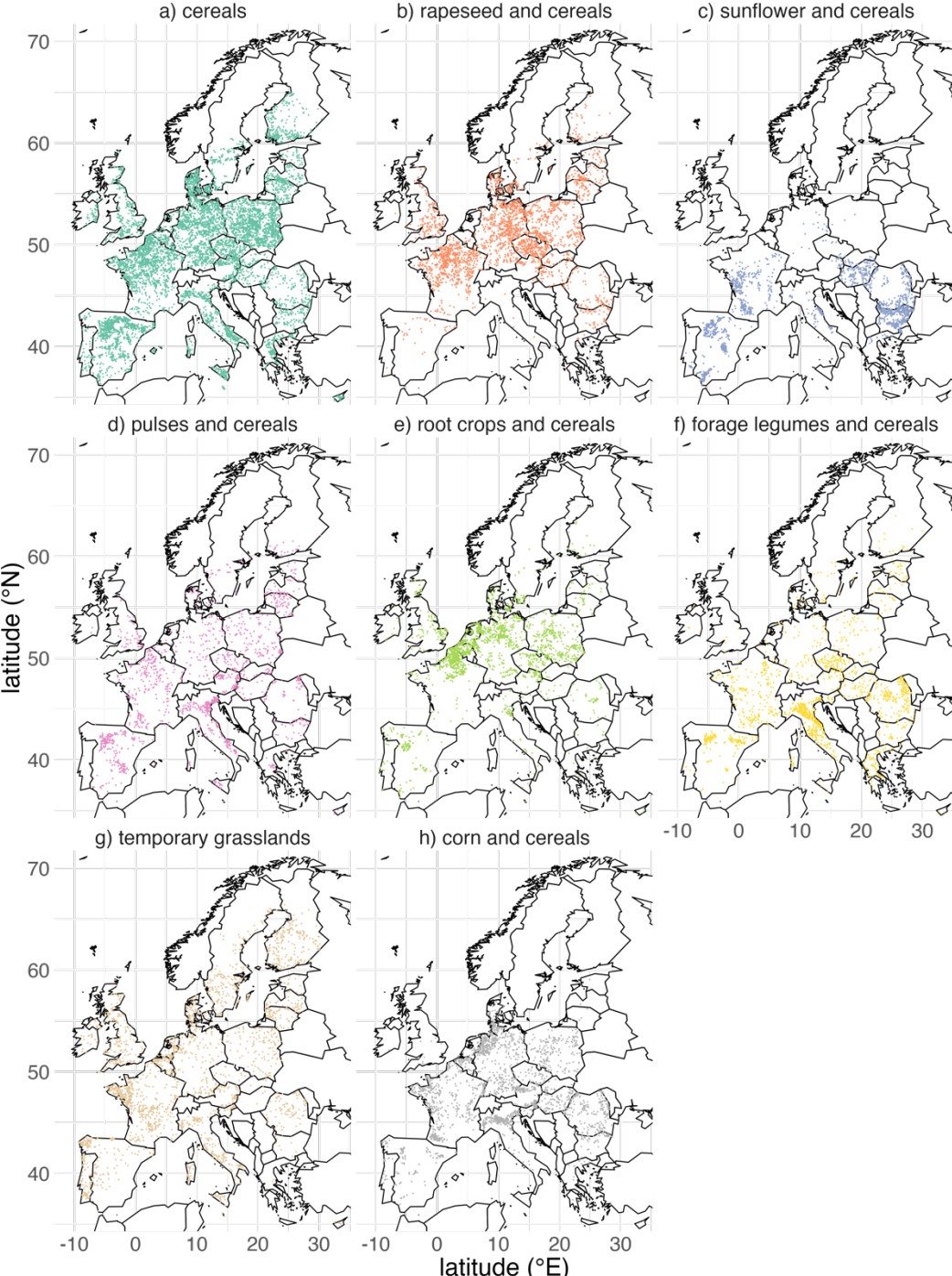

**Figure 8. Individual maps of crop sequence types in Europe based on LUCAS dataset.** Note that crop sequence types have been given short names that reflect dominant crops or group of crops within the sequence. For example, the crop sequence type "grasslands" also includes cereals and corn, but these crops are less frequent in the sequence. Composition of each crop sequence type in terms of temporal frequencies of (groups of) crops is presented in Table 2.

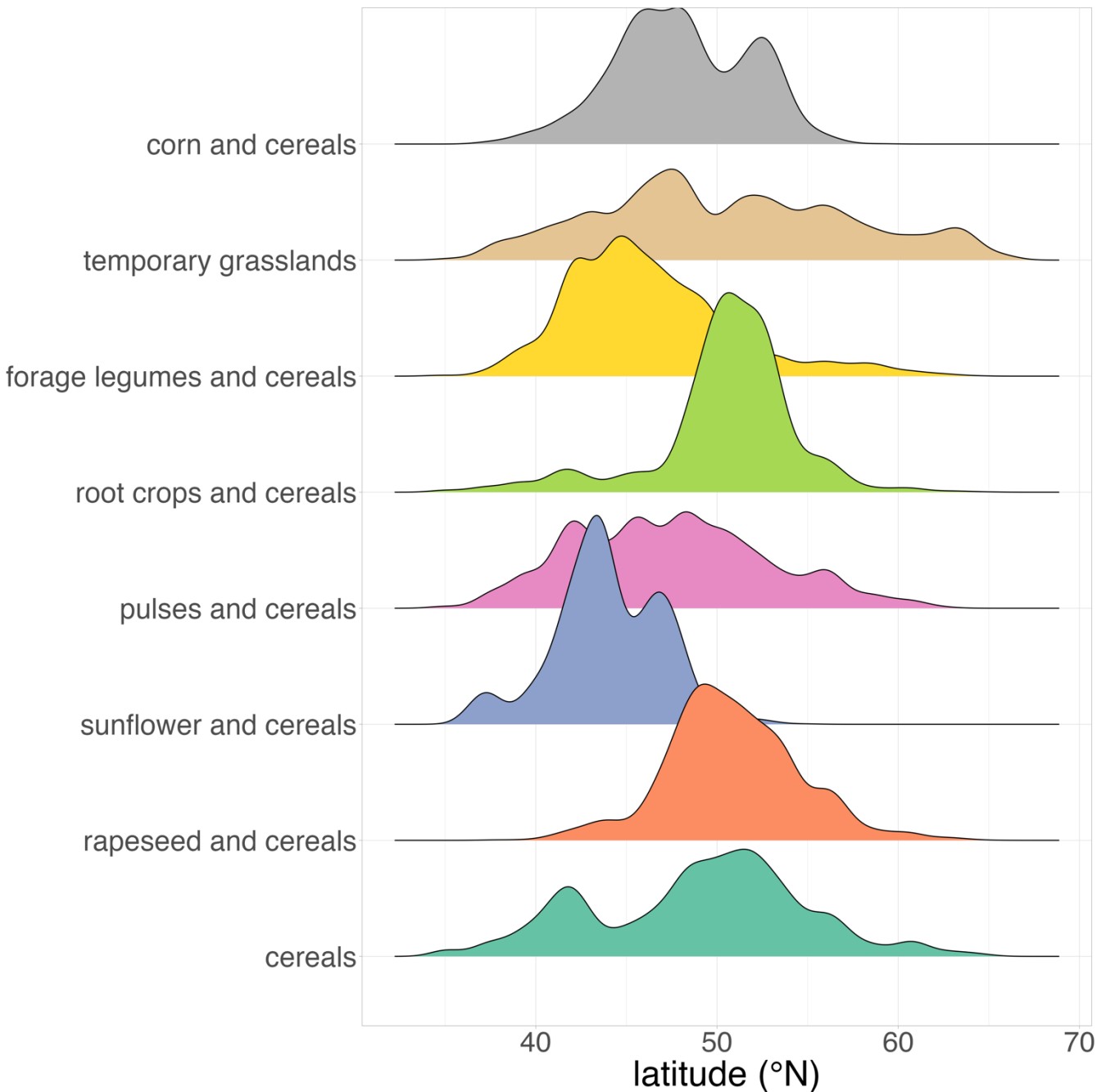

**Figure 9. Latitudinal distribution of crop sequence types based on the LUCAS dataset.** Note that crop sequence types have been given short names that reflect dominant crops or group of crops within the sequence. For example, the crop sequence type "grasslands" also includes cereals and corn, but these crops are less frequent in the sequence. Composition of each crop sequence type in terms of temporal frequencies of (groups of) crops is presented in Table 2.

**Table 1. Summary of data sources used in this study.**

| Name | Dataset description | References |
|---|---|---|
| **Harmonised LUCAS dataset** | Harmonised LUCAS *in-situ* land cover and use database for field surveys from 2006 to 2018 in the European Union | (d'Andrimont et al., 2020) Available for download at: https://data.jrc.ec.europa.eu/dataset/f85907ae-d123-471f-a44a-8cca993485a2 |
| **French LPIS dataset** | RPG Explorer Crop successions France 2007-2014, 2007-2019, 2015-2019 | Martin et al. (2021) Available for download at: https://data.inrae.fr/dataset.xhtml?persistentId=doi:10.15454/XH84QB |
| **Eurostat dataset** | Annual crop acreage at national scale for EU countries from 2000 | Available for download at: https://ec.europa.eu/eurostat/estat-navtree-portlet-prod/BulkDownloadListing?file=data/apro_cpsh1.tsv.gz |

530

**Table 2: Grouping of land cover categories considered for crop sequence classification from harmonised LUCAS dataset, and correspondence with French LPIS and EUROSTAT datasets used for validation**

|  | Examples of crops | Harmonised LUCAS dataset | French LPIS dataset | EUROSTAT dataset |
|---|---|---|---|---|
| Cereals | wheat, barley, oat, triticale, rye | LC B11 to B15 and B17 to B19 | ble, orge, cer_2nd | C1110 to C1420 and C1600 to C1900 |
| Corn |  | LC B16 | mais | C1500, G3000 |
| Rapeseed |  | LC B32 | colza | I1110 |
| Sunflower |  | LC B31 | TRN | I1120 |
| Pulses | dry pulses, soybean, | LC B33 and B41 | olea, prot | I1130, P110 to P9000 |
| Root crops | beets, potatoes | LC B21 to B23 | leg_fl, a_indus | R1000 to R3000 |
| Forage legumes | alfalfa, clover | LC B51 and B52 | fou | G2000 |
| Temporary grasslands |  | LC B53, B55 and E20 | pt, pp | G1000 |

**Table 3. Average temporal frequency of crops or crop groups within each crop sequence type.** Values in bold indicate the dominant (or two dominants) crop(s) or group(s) of crops in the crop sequence type. Values in brackets indicate standard deviation.

| Crop sequence type (short name) | Temporal frequency of crop or crop group over the 2012-2018 period | | | | | | | | | Relative importance of crop sequence types across EU |
|---|---|---|---|---|---|---|---|---|---|---|
| | Cereals | Corn | Rapeseed | Sunflower | Pulses | Root crops | Grassland | Forage legumes | TOTAL | |
| 1- Temporary grassland | 0.13 | 0.11 | 0 | 0 | 0 | 0 | **0.74** | 0 | 0,98 | 11% |
| | (0.16) | (0.22) | (0) | (0) | (0) | (0) | (0.20) | (0) | | |
| 2- Forage legumes and cereals | **0.23** | 0.11 | 0.02 | 0.02 | 0 | 0 | 0.13 | **0.47** | 0,98 | 7% |
| | (0.25) | (0.19) | (0.09) | (0.08) | (0) | (0.02) | (0.22) | (0.20) | | |
| 3- Corn and cereals | **0.17** | **0.82** | 0 | 0 | 0 | 0 | 0 | 0 | 0,99 | 10% |
| | (0.17) | (0.17) | (0) | (0) | (0) | (0) | (0.10) | (0) | | |
| 4- Root crops and cereals | **0.35** | 0.09 | 0.05 | 0 | 0 | **0.43** | 0.04 | 0.01 | 0,97 | 10% |
| | (0.26) | (0.18) | (0.13) | (0) | (0) | (0.19) | (0.13) | (0.05) | | |
| 5- Rapeseed and cereals | **0.43** | 0.09 | **0.43** | 0 | 0 | 0 | 0.03 | 0 | 0,98 | 13% |
| | (0.25) | (0.18) | (0.19) | (0) | (0) | (0) | (0.12) | (0) | | |
| 6- Sunflower and cereals | **0.39** | 0.11 | 0.03 | **0.42** | 0 | 0.01 | 0.02 | 0 | 0,98 | 8% |
| | (0.25) | (0.20) | (0.11) | (0.16) | (0) | (0.05) | ((0.10) | (0) | | |
| 7- Pulses and cereals | **0.36** | 0.09 | 0.04 | 0.03 | **0.37** | 0.03 | 0.04 | 0.01 | 0,97 | 7% |
| | (0.26) | (0.19) | (0.13) | (0.10) | (0.12) | (0.10) | (0.12) | (0.07) | | |
| 8- Cereals | **0.79** | 0.08 | 0 | 0 | 0 | 0 | 0.06 | 0 | 0,93 | 35% |
| | (0.16) | (0.13) | (0) | (0) | (0) | (0) | (0.11) | (0) | | |

**Table 4. Confusion matrix of the Random Forest model**

| Observed class | Predicted class | | | | | | | | Error rate (%) |
|---|---|---|---|---|---|---|---|---|---|
| | **1** | **2** | **3** | **4** | **5** | **6** | **7** | **8** | |
| 1- Cereals | 7564 | 0 | 0 | 0 | 0 | 0 | 0 | 0 | 0 |
| 2- Rapeseed and cereals | 0 | 2818 | 0 | 0 | 0 | 0 | 0 | 0 | 0 |
| 3- Sunflower and cereals | 0 | 0 | 1632 | 0 | 0 | 0 | 0 | 0 | 0 |
| 4- Pulses and cereals | 0 | 0 | 0 | 1504 | 0 | 0 | 0 | 0 | 0 |
| 5- Root crops and cereals | 0 | 0 | 0 | 0 | 2172 | 0 | 0 | 0 | 0 |
| 6- Forage legumes and cereals | 0 | 0 | 1 | 0 | 0 | 1478 | 0 | 0 | 0,07% |
| 7- Temporary grasslands | 0 | 0 | 0 | 0 | 0 | 0 | 2292 | 0 | 0 |
| 8- Corn and cereals | 0 | 0 | 0 | 0 | 0 | 0 | 0 | 2159 | 0 |

540

**Table 5: Spatial sampling effort of LUCAS data by country.** The sampling effort is quantified as the number of LUCAS points for which observations are available in 2012, 2015, and 2018 per unit area of arable land in 2018 by country as reported by (EUROSTAT, 2022)

| Country | Number of LUCAS points | Arable land area (x1000ha) | Number of LUCAS points / agricultural area |
|---|---|---|---|
| Montenegro | 12 | 8 | 1,42 |
| Cyprus | 84 | 94 | 0,89 |
| Slovenia | 83 | 167 | 0,50 |
| Luxemburg | 25 | 62 | 0,40 |
| Austria | 544 | 1408 | 0,39 |
| Belgium | 332 | 976 | 0,34 |
| Czech republic | 690 | 2069 | 0,33 |
| Italy | 1833 | 6142 | 0,30 |
| Latvia | 309 | 1075 | 0,29 |
| The Netherlands | 332 | 1269 | 0,26 |
| Finland | 565 | 2171 | 0,26 |
| Denmark | 601 | 2347 | 0,26 |
| Germany | 3007 | 11772 | 0,26 |
| Poland | 2448 | 10908 | 0,22 |
| France | 3988 | 19092 | 0,21 |
| Sweden | 507 | 2461 | 0,21 |
| Bulgaria | 668 | 3289 | 0,20 |
| Estonian | 118 | 661 | 0,18 |
| Spain | 2140 | 12396 | 0,17 |
| Ireland | 79 | 476 | 0,17 |
| Lithuania | 362 | 2209 | 0,16 |
| Greece | 320 | 1972 | 0,16 |
| Portugal | 169 | 1158 | 0,15 |
| Slovakia | 178 | 1358 | 0,13 |
| Romania | 1048 | 8946 | 0,12 |
| United Kingdom | 749 | 7023 | 0,11 |
| Hungary | 429 | 4228 | 0,10 |