# Peer review of "Figure S1. Comparison of the relative importance of crop sequence types derived from the LUCAS and French LPIS datasets at the national scale in France"

_Earth System Science Data, 2022_

## Author Response (AR1)

Dear Topical Editor,

Please find bellow a point-by-point reply to referees' comments.

We hope the improvements proposed in the revised manuscript will address all issues that were raised.

Sincerely your,

Rémy Ballot, on behalf of the authors
* * *
*Referee 1*

**Major comments**

Comment: *The manuscript proposes an interesting approach to map the crop rotation at EU-28 level by reconciling LUCAS surveys happening every three years and annual dataset from farmer's declaration. While the approach is promising and the authors provided interesting statements along with open source code, major improvements and clarifications are needed before being considered for publication.*

Response: We thank the reviewer for this positive feedback on our work.

Comment: *First of all, there is an inherent flaw in determining crop rotation from a 3 year (LUCAS) observation set and then comparing it to a subset (France) of annual rotations, and extrapolate that comparison to the EU. For instance, a common 3 year rotation like cereals - sugar beet - potatoes (e.g. in NW Europe) will only appear as a sequence of a single crop class in LUCAS, while being explicit in the annual France data set. This can only be resolved (for France) by comparing 3-year cycles starting each year (e.g. 2012 - 2015 - 2018, 2013 - 2016 - 2019 and 2014 - 2017 - 2020) and then aggregate co-located sequences into a LUCAS-like indicator. There is no discussion on this anywhere in the paper. This should be deeply clarified and discussed.*

Response: We agree with the reviewer on this point: "are temporally-incomplete crop sequences derived from LUCAS a good predictor of true (i.e. temporally-complete) sequences?" is a key question. However, we believe this is already addressed in the manuscript (see below), so that we disagree with the statement "There is no discussion on this anywhere in the manuscript". Indeed:

in lines 62-66 of the revised manuscript it is written "The purpose of this work is to map current dominant crop sequences from the European Land Use Cover Area frame statistical Survey (LUCAS). For this study, the multi-year harmonised data by d'Andrimont et al. (2020) was used. As this dataset is temporally incomplete (observations of land use on fixed points, only every three years), we proceeded in three steps (Figure 1) to assess how this incomplete information could be used to describe the diversity and localization of major crop sequences across Europe."

a full paragraph in the methods section (section 2.3.3 "Quality assessment") is dedicated to the explanation of how the comparison was performed. See especially lines 145-146 in revised manuscript "The crop sequence types derived from incomplete temporal sequences (LUCAS dataset) were compared to crop sequence types derived from complete temporal sequences (French LPIS dataset). To this aim, a three-step procedure was followed."

a full paragraph in the result section provides details on the comparison. See line 188 in revised manuscript section "3.2.1 Incomplete temporal crop sequences are a good proxy of complete temporal crop sequences at national and regional scale in France"

the limits of our approach based on LUCAS data in capturing the full diversity of crop sequences is already discussed in lines 261-263 in revised manuscript "A possible explanation for this overestimation could be the inability of the three sample years available in the LUCAS dataset (i.e. 2012, 2015, 2018) to capture the full diversity of longer crop rotations, such as rotations including temporary grassland. (and following sentences)."

The reviewer also states that "there is an inherent flaw in determining crop rotation from a 3 year (LUCAS) observation set and then comparing it to a subset (France) of annual rotations". The main reason given to support this statement is "For instance, a common 3 year rotation like cereals - sugar beet - potatoes (e.g. in NW Europe) will only appear as a sequence of a single crop class in LUCAS, while being explicit in the annual France data set". We disagree this a flaw in our approach. Instead, we do recognize temporally-incomplete sequences might not be a good proxy for temporally complete sequences for a number of reasons, including the one cited above by the reviewer, but this has to be tested and this is the purpose of our paper. We highlight that the issue raised by the reviewer about the common 3 year rotation like cereals - sugar beet – potatoes is not supported by our results as explained in lines 190-192 in the revised manuscript "On average at the national scale in France, the relative importance of crop sequence types estimated from the LUCAS dataset is in good agreement (R²=0.93, RMSE=0.04) with estimates based on the French LPIS dataset, with no systematic bias (Figure 3, Figure S1)." A possible explanation is that the possible "flaw" mentioned by the reviewer would become important only when all survey points are synchronized in terms of crop sequence, which is likely not the case. We add a few sentences in the discussion section to make this more explicit, lines 252-257 in the revised manuscript "This deserved to be tested, as numerous reasons could have led to inconsistent results. For example, a three-year cereal-beet-potatoes rotation, which is quite common in North-Western Europe, will appear in LUCAS dataset as cereal-cereal-cereal or beet-beet-beet or potato-potato-potato, depending on the crop present when observation is carried out. Thus, it will be classified as "cereals" or "root crops and cereals" crop sequence type. It did not affect the estimation of crop sequence types relative importance, as all survey points are not likely synchronized in terms of crop sequence."

The exact meaning of a "LUCAS-like indicator" for French LPIS data is not very clear, so that we did not make any change here.

*Comment: The groups of crop sequences (crop sequence types) identified by the PCA should be described in detail (which crops, how many of each of them) in the method part. This comes only in the result part (Table 2) and thus makes the understanding of the follow up of the methods very difficult as this result of the clustering is not described properly before.*

Response: From our point of view, the groups of crop sequences are a major result of our study and should remain in the results section. If the editor think it is necessary, we can move section 3.1 into section 2.3.2.

*Comment: Another issue is the decision of the authors to include permanent grassland (E20) in the temporary grassland class. Especially as in the results they show that this is the predominant class (Table 2) and that "LUCAS dataset overestimate the proportion of the "grasslands" crop sequence". This should be clarified in the manuscript.*

Response: Not all permanent grassland were included in the temporary grassland class, but only LUCAS points identified as permanent grassland in 2018, but in non-perennial agricultural use in 2012 and 2015. This is already written explicitly in lines 112-115 in the revised manuscript of the manuscript "As an exception, we also conserved points identified as permanent grasslands in 2018 (i.e. LC E20), but identified as a non-permanent agricultural use in 2012 or 2015, as they may be the result of a confusion between temporary and permanent grassland during observation.".

However we agree this might impact our results, and we modified the text in discussion section of the manuscript to further discuss the consequences of this choice on our results, lines 267-271 in the revised manuscript "In LUCAS dataset, we decided to consider as temporary grasslands points observed as permanent grassland in 2018, but as non-perennial agricultural cover in 2012 and 2015. These points are more likely temporary grasslands confused with permanent ones. But they could also be actual permanent grasslands, after a perennial change in land use between 2015 and 2012, leading to an overestimation of crop sequence with temporary grasslands importance."

*Comment: RPG is a so-called block system (a particular implementation of a LPIS for use in the CAP). A block can enclose several (1 or more) agricultural parcels with different crop types. Thus, an RPG parcel does not necessarily link to a single crop type. In earlier RPG use (e.g. 2012) the agricultural parcel was only indicated as an area estimate without specific geolocalization within the RPG parcel. Also, RPG has undergone significant change in this period, replacing the use of outdated cadastral parcels with a block system. This progress is regionally specific in France. This all makes spatially and temporally consistent point comparison difficult. It is relatively easy to highlight such issues in a graphical analysis. It is essential to include this analysis in the paper and discuss possible impact.*

Response: This is true regarding the raw RPG data, for the time period 2007-2014 (which overlaps the 2012-2018 we considered in our study). We do not use this raw data in our study, but the RPG explorer crop successions dataset provided by Martin et al. (2021). To develop this dataset, the authors used the RPG explorer algorithm, which follows a set of decision rules to identify the most likely crop sequence when there is more than one crop type in a given block. To make this more explicit, we elaborated the presentation of the dataset of Martin et al. (2021) in the method section, lines 88-91 in the revised manuscript "Until 2014, information was collected at the block scale. Each block can enclose one or more agricultural parcels, and thus one or more crops with declared area for each one but no geolocalisation within the block. Due to parcels reconfiguration from one year to another, it is not straightforward to know pluriannual crop sequences from the French LPIS, for years older than 2015." and 94-95 "It also relies on algorithm, which identifies the most-likely crop sequence when more than one crop is declared for one given block."

We also added a sentence in the discussion section about possible limitations of this dataset, lines 273-277 in the revised manuscript "The RPG explorer algorithm, which compiles LPIS annual raw data, into pluriannual crop sequence data, has been validated. However, it cannot be 100% correct, when identifying the most likely crop sequence, when more than one crop declared per block. A robust estimation of farmers' declaration error and how it could propagate into crop sequence is challenging and could not be done in this study. But to date, the dataset we used is the most complete regarding crop sequences knowledge in France."

*Comment: Crop attributes in RPG are not the same as (aggregated) crop class in the LUCAS nomenclature. The paper states that more than 300 crop types are included in the most recent RPG. There was likely some translation step to map 300 crop types into the N LUCAS groups. This should be explained. The problem with harmonizing at EU level is not so much related to spatial or temporal coherence (all national sets are large scale ortho- photo based, full territory and annual) but with consolidating the parcel attributes (reference to TUM effort).*

Response: We thank the reviewer for highlighting this point and give us the opportunity to clarify our method. We added a new table (Table 2) with crop names correspondence between LPIS, LUCAS and Eurostat datasets.

*Comment: Farmer declared crop attributes are not 100% correct (otherwise, there would be no need for controls). A key factor is the quality of the LPIS and the registration process. While that quality has improved significantly in the period 2012 - 2018, it is still prudent to expect a 2-4% "material error" in the data even in 2018. Again, it would be important to understand how errors propagate into the rotation pattern results.*

Response: We thank the reviewer for raising this point. We agree this is relevant, but a robust estimation of this error is challenging (as well as it propagation) and out of the scope of this study. Therefore we added a sentence in discussion section about LPIS potential errors, due to farmers' declaration, lines 272-273 in the revised manuscript "Reliability of our validation dataset also needs to be discussed. The French LPIS dataset is based on farmers' declarations, which are not 100% correct."

*Comment: In section 3.2, relative importance of crops are compared between LUCAS and LPIS. LUCAS is designed to be used as a regressor estimator of area at NUTS2 level. Therefore such comparison of*

*looking at the distribution of the occurrence of LUCAS points VS LPIS has a lot of caveats. It would be better to compare the area from the LPIS with the area estimates from LUCAS point.*

Response: The first part of this comment might be based on a misreading of our paper. Indeed, in section 3.2.1., we do not compare LUCAS and LPIS regarding importance of crops, but regarding the relative importance of crop sequence types, as already written in lines 190-192 of the revised manuscript "On average at the national scale in France, the relative importance of crop sequence types estimated from the LUCAS dataset is in good agreement ($R^2$=0.93, RMSE=0.04) with estimates based on the French LPIS dataset, with no systematic bias (Figure 3, Figure S1)".

Then, if we understand well, the suggestion here is to compare crop importance in terms of areas derived from both datasets. If this is correct, this is already done at Europe scale in section 3.2.2., e.g in lines 210-211 "Comparison of crop harvested areas derived either from crop sequence types or from official statistics (EUROSTAT, 2022) shows good agreement at the EU scale (Figure 5), with R2 higher than 0.98 and no bias. »

Finally, the reviewer did not explain what would be the caveats when comparing the "occurrence of LUCAS points VS LPIS", and it is not very clear if this comment refers to the comparison of relative importance of crop sequence types or to the comparison of crop harvested area. Therefore, we did not make any change here.

*Comment: We encourage the authors to make the improvements proposed to improve their interesting manuscript and we are providing further minor comments below.*

Response: Thank you for this positive feedback and for giving us the opportunity to clarify the manuscript. We hope the improvements proposed in the revised manuscript will address all issues that were raised.

**Minor comments**

*Comment: Language use is somewhat peculiar. Native English review would be beneficial.*

Response: From our point of view, the language is correct. We leave it to the editor to decide whether or not an English review is necessary.

*Comment: Montenegro is not an EU Member State. Remove all references to it, including the discussion on the low LUCAS point density.*

Response: We thank the reviewer for highlighting this point. We prefer to keep working with the whole harmonized LUCAS dataset. Indeed, this dataset contains information for the Great Britain that is no longer an EU member state, but no information for Croatia and Malta, which are actual EU member states. To ensure clarity and consistency, we rephrased the text where relevant to talk about "Europe" instead of the European Union for the spatial scope of this study. We also added a sentence describing the set of countries considered in this study, lines 81-84 in the revised manuscript "All EU-28 member states are represented in this dataset, except Croatia and Malta. Montenegro, which is not a member state, is also represented, as well as Great Britain, which is no longer a member state. Thus, "Europe" will be use hereafter to refer to the spatial scope of this study."

*Comment: Abstract and line 21: use of "essential linchpin" is odd. Probably "key element" is meant. It is "assumed to be a key element", there is not so much evidence that it actually is.*

Response: We replaced "essential linchpin" by "key element" as suggested by the reviewer. However, we believe that the evidence base regarding the role of crop diversification in agroecological transition is strong enough to keep "crop diversification is considered as key element" and not "crop diversification is assumed to be a key element". See references below:

Beillouin, D., Ben-Ari, T., Malézieux, E., Seufert, V., & Makowski, D. (2021). Positive but variable effects of crop diversification on biodiversity and ecosystem services. Global Change Biology, 27(19), 4697-4710.

Tamburini, G., Bommarco, R., Wanger, T. C., Kremen, C., Van Der Heijden, M. G., Liebman, M., & Hallin, S. (2020). Agricultural diversification promotes multiple ecosystem services without compromising yield. Science advances, 6(45), eaba1715.

Beillouin, D., Ben-Ari, T., & Makowski, D. (2019). Evidence map of crop diversification strategies at the global scale. Environmental Research Letters, 14(12), 123001.

Isbell, F. (2015). Agroecology: agroecosystem diversification. Nature plants, 1(4), 1-2.

Renard, D., & Tilman, D. (2021). Cultivate biodiversity to harvest food security and sustainability. Current Biology, 31(19), R1154-R1158.

Comment: L 10: "temporally-incomplete" maybe mention here the LUCAS data years (2012,2015 and 2018) that were used in the study?

Response: LUCAS data years were added as suggested.

*Comment: L19: The Zenodo link (https://doi.org/10.5281/zenodo.7016986 ) provides only a png low quality map of the points and a CSV table. I would have expected to have a georefenced dataset.*

Response: The main dataset shared through this link is the CSV table, including LUCAS points coordinates with crop sequence type associated. This dataset is accompanied by an illustrative map. If the editor think it is necessary, we could convert these files into a georeferenced dataset.

*Comment: L 21: Crop diversification is the process (action) that leads to crop diversity (status). Check the review paper by Hufnagel et al., 2020. They note that diversification is interpreted and defined differently in the scientific literature. Hufnagel et al define: "Crop diversification can be considered as an attempt to increase the diversity of crops through, e.g. crop rotation, multiple cropping or intercropping compared to specialized farming with the aim to improve the productivity, stability and delivery of ecosystem services". In the scientific literature, the most studied aspect of diversification is crop rotation/intercropping.*

Response: We agree with the reviewer. We made no changes here as no modification were requested.

*Comment: L31-34: Maybe add also this reference from Bohan et al. "Designing farmer-acceptable rotations that assure ecosystem service provision in the face of climate change"*

Response: We thank the reviewer for bringing to our attention this very interesting reference. However, lines 37-40 in the revised manuscript refer to foresight studies, i.e. studies investigating food systems at large scale (European Union, global scale). As the topic of Bohan et al (2021) is different from food systems modeling, we prefer not to add this reference here.

*Comment: L 43: harmonization of nomenclature is another issue*

Response: Thanks for pointing this out. We rephrased the text to mention the lack of harmonization of nomenclature as another issue in addition to spatial and temporal resolution, line 48 in the revised manuscript "lack of harmonization (e.g. spatial and temporal resolution or nomenclature)".

*Comment: L 48: "European" should be changed to "EU"*

Response: As mentioned previously following the reviewer's suggestion, we kept "European" as the scope of our study does not fully correspond to the EU. Indeed, some EU-countries are missing in the LUCAS dataset (Croatia, Malta). On the other hand, data is available for Montenegro, as well as for the Great Britain, which is no longer part of the EU.

*Comment: L 51-52: Suggestion to rephrase and to add sentence: "The purpose of this work is to map current dominant crop sequences from the European Land Use Cover Area frame statistical Survey (LUCAS). For this study, the multi-year harmonised data by d'Andrimont et al. (2020) were used."*

*It is important to clarify that LUCAS is carried out by EUROSTAT and what you have used is the multi year harmonised dataset.*

*Response*: Thanks for this suggestion. We rephrased as suggested to improve clarity, lines 63-64 in the revised manuscript.

*Comment: L 54: "preprocessed" should be changed by "filtered"*

*Response*: Done.

*Comment: L 66: "from" should be "since"*

*Response*: Done.

*Comment: L 66: EU-28, Europe, European Union. Different wording in the manuscript. I would suggest using EU-28 everywhere in the manuscript to avoid confusion.*

*Response*: As explained above, we modified the text where relevant to use "Europe" consistently throughout the manuscript. As explained above, the scope of our study covers countries with data available in the harmonized LUCAS dataset, which does not fully correspond to the EU-28. Some EU countries are missing (Croatia and Malta) and data is available for non-EU countries (UK and Montenegro).

*Comment: L 74: "over the period 2012-2018" The RPG Explorer Crop provides 2007-2019 according to https://entrepot.recherche.data.gouv.fr/dataset.xhtml?persistentId=doi:10.1 5454/XH84QB*

*Response*: This is true that this dataset includes years from 2007 to 2019 when this manuscript was written (and also 2020 now). However, we thought it was more relevant to indicate the number of fields for the period 2012-2018 to be consistent with the harmonized LUCAS dataset years used in this study (2012-2018).

*Comment: L 77 : This section should be more detailed and should provide reference to the Table 1.*

*Response*: As required changes were not specified, we did not make any modification. However, we added a reference to Table 1.

*Comment: L 83: six-year (not seven). Also, I would not say "the three most recent campaigns" as there was a campaign this year in 2022.*

*Response*: We did not change "seven" for "six", as 2012-2018 is a seven-year time period (2012, 2013, 2014, 2015, 2016, 2017, 2018). We added "the three most recent campaigns available at time of the study", line 105 in the revised manuscript.

*Comment: L 85 : The following statement "and thus we do not consider older campaigns (i.e. 2006 and 2009) which may be outdated to represent current crop sequences" is not exactly the reason you explain afterwards at the end of the paragraph, I would rephrase for the sake of consistency.*

*Response*: There are two different reasons why we did not consider campaigns in 2006 and 2009: (i) these years are outdated to represent current crop sequences, AND (ii) limited number of countries / points with data available for 2006 to 2018 campaigns. To make it more explicit, we moved lines 107-111, just after line 107, in the revised manuscript.

*Comment: L 101: "the temporal frequencies over three years" is non-sense. The LUCAS time span is 6 years, with only 3 observations. What you really do is tabulate for each point the crop (group) occurrence sequence and then map the spatial pattern for the 8 most important ones.*

*Response*: Indeed, this is exactly what we did.

*Comment: The key weakness in this approach is that different sequences may actually relate to the same 3 year rotation (e.g. as in the example above: if one point shows wheat - wheat - wheat and the next closest point to it: potato - potato - potato, it may be exactly the same rotation, but just shifted one year). This is why the comparison to annual rotation in France is impossible to extrapolate to LUCAS-based rotation in the EU.*

Response: As explained previously, we agree with the reviewer that temporally-incompleteness of LUCAS data might lead to biased estimates of crop sequences as the reviewer describes above. But again, this claim is not supported by our results as described in lines 190-192 in the revised manuscript "On average at the national scale in France, the relative importance of crop sequence types estimated from the LUCAS dataset is in good agreement ($R^2$=0.93, RMSE=0.04) with estimates based on the French LPIS dataset, with no systematic bias (Figure 3, Figure S1)."

*Comment: L 105-18: presenting this as a table would ease the reading.*

Response: We added a table (Table 2) with correspondence between (group of) crops names, codes in LUCAS nomenclature and in LPIS nomenclature.

*Comment: L 108: "(viii) temporary grassland (LC B53, B55, and E20)" E20 is NOT a temporary grasslands. Check here p.58 https://ec.europa.eu/eurostat/documents/205002/8072634/ LUCAS2018-C3-Classification.pdf . "This class excludes Temporary grassland and fodder crops (B5X)". This has to be clarified*

Response: We agree that E20 is not temporary grassland. However, we already explained in lines 112-115 in the revised manuscript the reason why we considered some points classified E20 in 2018, but under non-perennial agricultural use in 2012 and 2015: "As an exception, we also conserved points identified as permanent grasslands in 2018 (i.e. LC E20), but identified as a non-permanent agricultural use in 2012 or 2015, as they may be the result of a confusion between temporary and permanent grassland during observation."

*Comment: L 113-116: The groups of crop sequences (crop sequence types) identified by the PCA should be described in more detail (which crops, how many of each of them)... This comes only in the result part (Table 2) and thus makes the understanding of the follow up of the methods very difficult.*

Response: From our point of view, the groups of crop sequences are a major result of our study and should remain in the result section. If the editor think it is necessary, we can move section 3.1 into section 2.3.2.

*Comment: L 120: the use of an RF model is poorly understood. The LUCAS sequences are directly given by the observation data, there is no need for any model to predict those.*

Response: The RF model is not used to predict crop sequence types in LUCAS dataset. It is trained on the LUCAS classified dataset and then applied to the French LPIS dataset to predict crop sequence type. This is already explained in Figure 1 and in lines 147-151 in the revised manuscript:" First, a random forest (RF) model was trained on LUCAS data to predict crop sequence type from crop (or crop group) frequencies (i.e. a total of eight predictors). The RF model was fitted using the randomForest() function of the R package randomForest v4.6.14 (Liaw and Wiener, 2002), with default settings. The RF model showed good performances, as indicated by an out-of-bag (OOB) error rate of only 0.12% (Table 3). Second, the RF model was applied on French LPIS data to classify observed crop sequences into the eight crop sequence types »

*Comment: L 123-224: applying the RF model trained on 3 year LUCAS observations to annual sequences is even less comprehensible, as the input to the RF inference is not of the same nature as the training. This is a very weak part of the paper, which would normally lead to rejection, as all the subsequent "patterns" are derived from it. As suggested above, the spatial pattern comparison between LUCAS vs France data should be derived from France data that is also spaced in 3 year intervals. This would also help to resolve how dominant regular rotations (e.g. 2, 3, 4 or 5 year rotation) are. Longer rotations are potentially of more interest in a biodiversity context (which is the "linchpin" etc.).*

*In light of the weakness of the approach, the results are likely less meaningful than proclaimed in the manuscript. They are more likely derived directly from the 3 year LUCAS (grouped) observation set, without the need for the RF model and "validation" with an annual data set. For sure, there is currently*

*little added value in the extrapolated results from the France set. This simplification could probably be addressed in a major revision of the paper.*

Response: We disagree with the statement "applying the RF model trained on 3 year LUCAS observations to annual sequences is even less comprehensible, as the input to the RF inference is not of the same nature as the training.". Indeed, the input to the RF model for inference and training are of the same nature, which is the temporal frequencies of groups of crops. The only difference is that those temporal frequencies are calculated over 3 years (out of 7 years) for the LUCAS data, and over 7 years (out of 7 years) for the French LPIS. Our results show that using 3 years out of 7 years to compute temporal frequencies of groups of crops is a good proxy of true temporal frequencies of groups of crops. We agree that rotations longer than 7 years would probably not be captured well by this approach, but we already discuss this in lines 261-266 in the revised manuscript "A possible explanation for this overestimation could be the inability of the three sample years available in the LUCAS dataset (i.e. 2012, 2015, 2018) to capture the full diversity of longer crop rotations, such as rotations including temporary grassland. For example, let's consider a cyclical crop rotation starting in 2012 with 3 years of consecutive grassland followed by wheat, maize and barley. Then, observation in 2012, 2015 and 2018 would be grassland, wheat, and grassland respectively, yielding a proportion of grassland of two-third instead of half.".

Regarding the temporal non-exhaustivity of LUCAS data, we think we can only describe crop sequences from it in terms of crop temporal frequencies, not in terms of rotation duration. We do not understand how the approach suggested here (deriving 3 years-spaced LUCAS-like data from French LPIS) could help to tackle rotation duration.

*Comment: L165: statement "LUCAS dataset overestimate the proportion of the "grasslands" crop sequence." should be reconsidered when taking into account temporary/permanent grassland properly in LUCAS.*

Response: See our answer about temporary/permanent grasslands above.

*Comment: L 380: Figure 1. Typo. "Average" and not "Avergae".*

Response: Done.

Referee 2

*Comment: On the positive side, I would mention the originality of the study based on rarely used data (LUCAS), and on a little used European scale. Also noteworthy is the representation of the latitudinal distribution of crop sequences, which is original (with the nuance that one would obtain globally the same result by representing only the heads of rotation of these crop sequences).*

Response: We thank the reviewer for this positive feedback on our work.

*Comment: This data-paper is of great methodological interest for learning about these LUCAS data, but it is difficult to convince about their usefulness for characterizing crop sequences. Indeed, temporal continuity seems essential to represent crop sequences and the LUCAS data are only surveyed every 3 years, which seems to be a huge bias for analyzing crop sequences that are essentially multi-year.*

*There is therefore a gap between what this study wishes to propose and what it actually provides. By redefining less ambitious objectives, it would gain in interest because it would not overestimate its potential.*

Response: We agree the initial version of the manuscript was somewhat confusing due to a gap between the title that mentioned "crop sequences" and the results that provided only estimates of crop temporal frequencies for 8 crop sequence types, and no exact order of succession of crops. We thank the reviewer for highlighting this issue. To improve clarity, we modified the title of the article to "The first map of crop sequences types in Europe over 2012-2018". Following comments that are made below, we also added a few sentences in the text to (i) provide a definition of "crop sequence", (ii)

better explain that the added value of our work is to get robust estimates of temporal crops frequencies in the sequence but that the exact order of crop succession could not be captured due to the nature of the LUCAS dataset, (iii) temporal crop frequencies are still a key feature of crop sequences with important agronomical relevance, (iv) capturing the exact order of succession would deserve further research. We hope these changes address the issue made by the reviewer. They are explained in details below.

**Major comments**

*Comment: The introduction focuses much of the article's interest on crop diversification. However, the results do not highlight diversification as such, but the distribution of the main "crop sequences", represented by their main rotation heads. In order to represent diversification, it would have been necessary to calculate a diversity indicator at the scale of relevant grid cells (country, NUTS3 etc.).*

Response: We understand the point raised by the reviewer. However, as explained in the text (see lines 35-36 in the revised manuscript), our main purpose was to develop a map of current dominant crop sequences (types) to be used as a baseline in foresight studies involving scenarios of crop sequence diversification. Therefore, our purpose was not to quantify the crop diversity within current crop sequences across countries or NUTS. We agree this is a relevant question, but we believe that calculating a diversity indicator as suggested would require more work and is out of the scope of this data paper. One reason underlying this point of view is the difficulty to reduce "crop diversity" to a single indicator, because different facets of the crop diversity are relevant to different ecosystem services. For example, the diversity of sowing dates is relevant to weed control (Weisberger et al., 2019), the crop functional diversity is relevant to nitrogen cycling when considering legume and non-legume crops (Zhao et al. 2022), and the phylogenetic crop species diversity might be relevant or not to the control of pests and diseases depending on whether considered crop species are hosts of the same pests and diseases or not (e.g. wheat and barley vs. wheat and soybean) (Beillouin et al. 2021). However, as said above, we agree this is a relevant question, and we added a few sentences in the discussion section to elaborate on this, lines 314-323 in the revised manuscript.

Beillouin et al. (2021). Positive but variable effects of crop diversification on biodiversity and ecosystem services. Global Change Biology, 27(19), 4697-4710.

Weisberger et al. 2019. Does diversifying crop rotations suppress weeds? A meta-analysis. PLoS ONE 14, 1–12. https://doi.org/10.1371/journal.pone.0219847

Zhao et al. (2022). Global systematic review with meta-analysis reveals yield advantage of legume-based rotations and its drivers. Nature Communications, 13(1), 4926.

*Comment: l.36: The concept of "crop sequence" is not defined here and this is detrimental to the rest of the presentation. Indeed, the authors should clarify what they mean by this term and why they chose this term rather than another (e.g. succession, rotation, or even a combination of majority crops) because if it means "an orderly sequence of crops on the same plot", it does not apply well to this study. Indeed, as it is stated below (l.39): "the LUCAS dataset, provide information about land use categories and crop species cultivated on agricultural land, but no information about crop sequences because data are not available every year. There is therefore a contradiction in the introduction because the purpose of this data-paper is to estimate crop sequences using these same data.*

Response: We thank the reviewer for giving us the opportunity to clarify this point. We added a definition of the term "crop sequence" along with a justification of why we chose this term rather than another, lines 30-36 in the revised manuscript "The term crop sequence refers to the sequence of crops grown in succession in the same field over a given period of time (Dury et al. 2012). A crop sequence is then defined by the nature of its crops and their order of succession. Based on this definition, the temporal frequencies of crops is a key feature of a crop sequence. The term crop rotation is also commonly used to refer to the sequence of crops grown in succession in the same field (Bullock, 1992), but includes a notion of cyclicality (e.g. rotation length) – at least to some degree

(Castellazzi et al. 2007). Hereafter, we use the term crop sequence rather than crop rotation because we consider a fixed period of time (i.e. 2012-2018), focus on temporal frequencies of crops, and do not analyze cyclicality in crop sequences."

Regarding the comment that the LUCAS dataset is not suited to analyze crop sequences, we agree that the initial text was confusing. To improve clarity, we rephrased the text in lines 53-59 in the revised manuscript as follows: "To overcome this problem, we developed an original method that combines European-level and national-level datasets to create a map of current dominant crop sequences at the European level. We show that the temporally-incomplete (i.e. every 3 years) information on crop sequences provided by the LUCAS dataset can be used to derive robust estimates of crops frequencies in the sequence when compared to crop frequencies derived from a temporally-complete national-level dataset. We acknowledge this approach does not allow capturing the exact order of crops in the succession. Nonetheless, it allows capturing crop frequencies which are a key feature of crop sequences (Castellazzi et al. 2007; Peltonen-Sainio and Jauhiainen 2019)".

Bullock, D. G. (1992). Crop rotation. Critical reviews in plant sciences, 11(4), 309-326.

Castellazzi, M. S., Wood, G. A., Burgess, P. J., Morris, J., Conrad, K. F., & Perry, J. N. (2008). A systematic representation of crop rotations. Agricultural Systems, 97(1-2), 26-33.

Dury, J., Schaller, N., Garcia, F., Reynaud, A., & Bergez, J. E. (2012). Models to support cropping plan and crop rotation decisions. A review. Agronomy for sustainable development, 32, 567-580.

Peltonen-Sainio, P., & Jauhiainen, L. (2019). Unexploited potential to diversify monotonous crop sequencing at high latitudes. Agricultural systems, 174, 73-82.

*Comment: Indeed, since the LUCAS data are 3 years apart, it is impossible to capture the "patterns" of the crop sequences, only the occurrence or not of the main crops. The diversity of these crops is also greatly reduced by the groupings made and by the deletion of the "other crops" category, which would include all diversification crops (forage mixtures, industrial crops, etc.).*

Response: We agree with the reviewer: due to the nature of the LUCAS dataset (i.e. data every 3 years) our approach does not allow capturing the exact order of crops in the succession. To make this more explicit, we modified the text (see our answer to the previous comment), and added a sentence in the discussion.

We also agree with the comment "The diversity of these crops is also greatly reduced by the groupings made and by the deletion of the "other crops" category", and we thank the reviewer for point this out. We added a sentence in the discussion to highlight this, lines 307-308 in the revised manuscript "We acknowledge that the grouping of crops and classification of crop sequences we made greatly reduced the diversity of crop sequences."

*Comment: In the end, the typology that has been carried out comprises only 8 types characterized by 8 groups of rotation heads associated or not with cereals. In my opinion, these are more crop combinations than crop sequences as such. The essential question to be asked in discussion would be whether there is any real added value in this approach compared to a simple mapping of the crop rotations of these same (groups of) rotation heads. There is indeed a compromise to be found between too much complexity, which would make the results unreadable, and too much simplification, which only deals with the major trends in crop distribution but not with crop sequences, with notions of the time required for the crop to return to itself, the length of the succession, etc.*

Response: It is well known that temporal and spatial crop diversity are not independent from each other (Aramburu Merlos and Hijmans, 2020). Therefore, it is expected that a crop sequence type characterized by a high temporal frequency of a given (group of) crops, will be frequent where this (group of) crops is widely cultivated. For example, crop sequences including corn can only be observed where corn is grown. However, knowing where corn is grown does not tell anything about the crop sequence in which corn is cultivated. Of course, knowing which other crops are grown in the same area

than corn can inform about possible crop sequences, but this is not sufficient. Our results provide a good example: many crop sequence types coexist in the very north of France (temporary grasslands, corn and cereals, root crops and cereals, rapeseed and cereals, pulses and cereals and cereals). As a consequence, cereals (e.g. wheat) can be found in very specialized crop sequences (e.g. the "cereals" crop sequence type), moderately diversified crop sequences (e.g. the "root crops and cereals" crop sequence type) and diversified crop sequences (e.g. the "pulses and cereals" crop sequence type). This demonstrates that specialized crop sequences can still occur in areas where a substantial diversity of crops is cultivated, and this cannot be inferred from land use (e.g. individual crop maps of "rotation heads" as suggested) only. We modified the text in lines 233-245 in the revised manuscript to better explained the added value of our approach.

Aramburu Merlos, F., & Hijmans, R. J. (2020). The scale dependency of spatial crop species diversity and its relation to temporal diversity. Proceedings of the National Academy of Sciences, 117(42), 26176-26182.

*Comment: l.54: I would therefore not use the term "diversity of crop sequences" but rather "localisation of major crop sequences*

Response: We changed to "the diversity and localization of major crop sequences".

*Comment: l.94: This bibliographic section explains well why it is impossible to study crop sequences with LUCAS: a large part of the crop sequences being over 3 years, LUCAS systematically misses them. The data-paper should be adapted to what LUCAS can really be used for, i.e. estimating crop frequencies at the scale of the surveyed point, which can then be aggregated to the level desired to estimate crop diversity.*

Response: As said previously, we agree the initial text was confusing as it mentioned "crop sequences" whereas our results only captured crop frequencies and not the exact order of succession due to the nature of the LUCAS dataset. As mentioned before, we added a definition of the term "crop sequence" along with a justification of why we chose this term rather than another, and we rephrased the text in lines 53-59 to better explain that we only analyze crop frequencies (that are still a key feature of crop sequences with important agronomical relevance).

*Comment: 108: In what categories are all the other crops (field vegetables, industrial crops, forage mixes, etc.)?*

Response: We thank the reviewer for highlighting that we forgot to explain how other crop categories were handled. Except industrial crops, they are in none of the eight categories considered. For a given point, the sum of frequencies calculated for the eight categories may be lower than one, the difference corresponding to the other crops. We added a sentence in lines 132-133 in the revised manuscript to explain this point "As these eight groups did not encompass all the land cover categories, the sum of (groups of) crops frequencies may be lower than 1 for each points.", and modified Table 2 to make it explicit.

*Comment: 123: the second step would deserve more explanation: in what form are the LPIS data presented? are they sequences of crops over 6 years? show examples of groupings...*

Response: LPIS data is presented as sequence of crops over 7 years. We will add this explanation along with examples. We changed the title of section 2.3.1. into "Preprocessing of LUCAS and French LPIS data" and added a sentence at the end of this section to make it explicit, that we calculated crop frequencies from sequences of crops, lines 134-135 in the revised manuscript "In order to serve the quality assessment step, the French LPIS dataset was preprocessed the same way. First, fields under perennial use were discarded. Then, (groups of) crops frequencies were calculated for each fields.".

*Comment: 165: specify that there is a 12-point difference between the two sources. This should lead to questioning the choices that were made to keep permanent grasslands considered as "temporary grasslands" in the "Preprocessing of LUCAS Data" section and retest the method.*

Response: We modified the text as suggested to specify the overestimation of +11% for grasslands: "However, estimates based on the LUCAS dataset overestimate the proportion of the "grasslands" crop sequence type (+11%), and slightly underestimate the proportion of the "rapeseed and cereals" (-5%), "root crops and cereals" (-3%), and "forage legumes and cereals" (-2%) ones", lines 192-194 in the revised manuscript.

As proposed in reply to a comment from referee 1, we elaborated in the discussion section on the way permanent grasslands were handled. We highlight that we do not consider all permanent grasslands as temporary grasslands, but only points identified permanent grassland in 2018, but identified as non-permanent agricultural use in 2012 and 2015, which are obviously not permanent grasslands.

*Comment: l.207: the last sentence "This demonstrates the importance of considering crop sequences in addition to land use, which do not reflect this diversity" is a bit excessive insofar as the same trends would be observed with land use data.*

Response: We rephrased the sentence to moderate the affirmation. However, the information is not quite the same than what could be observed with land use data. For example, a diversified land-use for a given region could reflect either crop sequence including a large diversity of crops on all fields, or contrasted but less diversified crop sequences on different subsets of the region, see lines 233-245 in revised manuscript "It is well known that temporal and spatial crop diversity are not independent from each other (Aramburu Merlos and Hijmans, 2020). Therefore, it is expected that a crop sequence type characterized by a high temporal frequency of a given (group of) crops, will be frequent where this (group of) crops is widely cultivated. For example, crop sequences including corn can only be observed where corn is grown. However, knowing where corn is grown does not tell anything about the crop sequence in which corn is cultivated. Of course, knowing which other crops are grown in the same area than corn can inform about possible crop sequences, but this is not sufficient. Our results provide a good example: many crop sequence types coexist in the very north of France (temporary grasslands, corn and cereals, root crops and cereals, rapeseed and cereals, pulses and cereals and cereals). As a consequence, cereals (e.g. wheat) can be found in very specialized crop sequences (e.g. the "cereals" crop sequence type), moderately diversified crop sequences (e.g. the "root crops and cereals" crop sequence type) and diversified crop sequences (e.g. the "pulses and cereals" crop sequence type). This demonstrates that specialized crop sequences can still occur in areas where a substantial diversity of crops is cultivated, and this cannot be inferred from land use (e.g. individual crop maps of "rotation heads") only."

**Minor comments**

*Comment: l.19: the legend would have deserved more contrasting colors to better highlight the different categories.*

Response: We are not sure which figure this comment refers to? Nevertheless, we highlight that on the map in Figure 7 (which is available in Zenodo) we used a colorblind-friendly palette to represent crop sequence types. We agree this map might be somewhat difficult to read, but maps of individual crop sequence types in Figure 8 should be helpful in this regard.

*Comment: l.55: give an example to clarify (ex. point X: Corn 33% Wheat 66%)*

Response: We added an example as suggested lines 68-69 in the revised manuscript "For example, a point identified as wheat in 2012 and corn in 2015 and 2018 was converted into: cereal frequency = 0,33, corn frequency = 0,67."

*Comment: 1.66: Be clear about the scope of the study: is it the EU with 27 or 28 members? In the table on p. 26, there are only 27 because Croatia and Malta are missing. On the other hand, Montenegro is not included, nor is Great Britain.*

Response: We agree we need to clarify this point, which has also been raised by the other reviewer. We decided to keep working with the full harmonized LUCAS dataset that includes some non-member

states (e.g. Montenegro, UK) while omitting some member states (e.g. Croatia, Malta). We therefore rephrased the text where relevant to use the word 'Europe' instead of "EU" or "European Union".

*Comment: L.74: Specify that there was another change in 2015: the grid used to represent land use changed in 2015 from the crop block to the plot, which greatly simplified the reconstruction of crop sequences.*

Response: We added this information (this was also requested by the other referee), lines 88-91 in the revised manuscript "Until 2014, information was collected at the block scale. Each block can enclose one or more agricultural parcels, and thus one or more crops with declared area for each one but no geolocalisation within the block. Due to parcels reconfiguration from one year to another, it is not straightforward to know pluriannual crop sequences from the French LPIS, for years older than 2015."

*Comment: L.78: Specify how this information is collected: e.g., are we sure that the European statistical services (EUROSTAT) do not use LUCAS to estimate these areas since they also manage LUCAS?*

Response: EUROSTAT is independent from LUCAS data, (see section 18.3 in https://ec.europa.eu/eurostat/cache/metadata/en/apro_cp_esms.htm). We modified the text to make this explicit, lines 99-101 in the revised manuscript "The crop statistics are collected by the National Statistical Institutes and/or Ministries of Agriculture by using several statistical methods: sample surveys, administrative sources, expert estimates. Most often a combination of several methods is used. Eurostat is independent from LUCAS data."

*Comment: l.86: Were only non-agricultural or permanent crop points in 2018 removed ("during the last observation")? What about points that were represented by permanent crops and/or non-agricultural occupations in 2012 and 2015?*

Response: We removed all points under non-agricultural use or permanent agricultural use in 2012, 2015, and 2018, except points identified as permanent grasslands in 2018 (i.e. LC E20), but identified as a non-permanent agricultural use in 2012 or 2015. We rephrased the text to make it more explicit, lines 111-112 in the revised manuscript: "We discarded points labelled as non-agricultural use or permanent agricultural use (e.g. orchards, vineyards) in at least one year among 2012, 2015 and 2018 (i.e. Land Cover (LC) not included in land cover classification B11 to B55)."

*Comment: l.91: Correct "2021" to "2012*

Response: Done.

*Comment: l.91: Is the perimeter 27 or 28 countries? To be harmonized with line 66.*

Response: See our previous answer on this point.

*Comment: l.230: Should Montenegro be kept in the analysis, given that it is outside the EU and has only 15 surveyed points?*

Response: See our previous answer on this point.

---

## Author Response (AR2)

Dear Topical Editor,

Please find bellow a point-by-point reply to referees' comments.

We hope the improvements proposed in the revised manuscript will address all issues that were raised.

Sincerely yours,

Rémy Ballot, on behalf of the authors

**Referee #2**

*Comment: L. 165 : To which figure do the references to Figures S1, S2 and S3 refer? I can't find them in the illustrations.*

Response: Figures S1, S2 and S3 are provided as supplementary information, which was uploaded as a separate file.

**Referee #3**

*Comment: The manuscript developed the first map of dominant crop sequences in Europe for 2012-2018 using temporally-incomplete land cover information. Obviously, this is an important work. I have reviewed the revised manuscript and the point-by-point responses to the comments from the other two reviewers. Overall, the authors have done a good job in addressing these comments and revised the manuscript. However, I still have concerns about the data processing and quality assessment.*

Response: We thank the referee for this positive feedback and hope the revisions proposed will address all concerns raised.

*Comment: 1. According to the description in Section 2.3.1, two crop sequences can be derived based on the LUCAS dataset in 2012, 2015, and 2018 for each field among the 31159 points (Line 116). So, the two crop sequences can be different in each field. How many points have different crop sequences?*

*In the following section, a principle component analysis (PCA) was performed and eight groups (crop rotations) were selected to represent the primary crop sequence. So, if one of two different crop sequences in a field doesn't exist in the eight groups, will it be discarded? If you process the dataset like this, it will result in some uncertainties.*

Response: For each point considered, a unique land cover is given for 2012, 2015 and 2018. Thus, a unique 2012-2015-2018 crop sequence is associated to each point. We added a sentence lines 116-117 in the revised manuscript to make this explicit: "Thus, each point is associated with a single (temporally incomplete) crop sequence.".

The PCA and hierarchical clustering performed aimed at distributing all points within the eight groups: no points were discarded here. We added a sentence lines 143-144 in the revised manuscript "A crop sequence type has thus been assigned to all points considered.".

Finally, line 114 in the revised manuscript, we specified how many points were added, identified as permanent grassland in 2018, but as a non-permanent agricultural use in 2012 or 2015. While adding this information and despite all previous verifications done, we identified a code mistake in lines 55-57 from the "1_LUCASD_formating.R" file. This mistake resulted in keeping in the analysis all points under permanent grassland in 2012, 2015, 2018. A corrected version of the script was shared on Zenodo and all relevant figures, tables and

manuscript sections were corrected accordingly. This change leads to drop one third of Lucas points previously considered and crop sequences with temporary grasslands were no longer overestimated, but crop sequence types characteristics remains stable, as well as conclusions of this work.

*Comment: 2. Dataset validation still needs to be improved for the manuscript. Though the authors have conducted comparisons analysis with LPIS and Eurostat datasets, the data accuracy outside of France has yet convinced me. Moreover, I did not see any new efforts to prove the reliability of the crop sequence in the revised manuscript. For example, they can collect crop sequence information from agricultural field studies to validate their dataset.*

Response: We agree the validation step is a critical step. However, collecting crop sequence information from agricultural fields as suggested to further validate our results may not provide a robust comparison because it would provide additional information about crop sequences cultivated on a limited number of fields, with a high risk of sampling bias. Moreover, for France, it would not improve the validation already performed by comparison with the LPIS, which provides an almost exhaustive knowledge about actual crop sequences in farmers' fields. We added a sentence lines 148-151 in the revised manuscript to strengthen this validation "The French LPIS represents the best available data for spatial distribution of crops in France in terms of spatial and temporal resolutions, spatial coverage, and disaggregation by crop type, , with a coverage higher than 98% for all field crops or 92% for temporary grasslands (Cantelaube et Lardot, 2022, Guilpart et al., 2022).". For other countries than France, we acknowledge that the validation could benefits from a comparison with datasets equivalent to the French LPIS : "the quality assessment of our map of crop sequence types in the EU would benefit from comparisons with other datasets at national or subnational levels in other countries" (lines 307-308 in the revised manuscript). But so far, we do not have access to such datasets.

*Comment: 3. Random Forest (RF) model was applied to evaluate the accuracy of the crop sequence map for France. It is critical for the quality assessment, so detailed explanations about the RF model should be included in section 2.3.3. For example, how many samples were selected to train and validate the RF model? How the model performs for the model training and validation? Table 4 shows the confusion matrix of the Random Forest model, but it needs to be clarified that the results were calculated based on training data or validation data.*

Response: We thank the reviewer for giving the opportunity to clarify this point. Default settings were specified line 154 in the revised manuscript : "i.e. 500 trees, two variables randomly sampled as candidates at each split". The out-of-bag error is a metric commonly used to evaluate random forests classifiers, and is considered to yield similar results than cross-validation. To make this more explicit in the text, we added the following sentence and reference in lines 155-159 of the revised manuscript: "Each tree of our RF model is constructed based on a random sample of the observations generated by bootstrap. The observations that are not part of the bootstrap sample are referred to as OOB observations, which are being used for estimating the prediction error, the so-called OOB error. The OOB error is considered as a good measure of the true prediction error (Matthew, 2011; Janitza and Hornung, 2018)."

*Comment: 4. It confused me that why the RF-predicted crop sequence map was used to compare rather than directly using the newly developed crop sequence map.*

Response: We changed the sentence lines 160-161 in the revised manuscript to make it clearer "the crop sequence type distributions derived from the LUCAS dataset and predicted from

French LPIS dataset were compared". In other words, comparison was made between the newly developed crop sequence types map from LUCAS on the first hand, and the RF-predicted crop sequence types map from LPIS on the other hand, as shown in Figure 1.

*Comment: 5. The newly developed crop sequence map is also compared with the Eurostat dataset by calculating their relative frequencies within the crop sequence types derived from the LUCAS dataset, the relative importance of crop sequence types, and the total arable land area. However, the newly developed crop sequence map is a point dataset, so how can the arable land area and the relative importance of each crop sequence type be calculated? Does the LUCAS data include field area information? I did check the Harmonized LUCAS data (https://developers.google.com/earth-engine/datasets/catalog/JRC_LUCAS_HARMO_THLOC_V1#table-schema), and I only found an attribute called "parcel_area_ha (i.e., Size of the surveyed parcel in hectares)". There is also no parcel area information in data shared on the Zenodo.*

Response: Relative importance was calculated regarding the number of points. We added an equation in lines 167-168 in the revised manuscript and justification in lines 171-172 "Relative importance of crop sequence types were calculated according to the number of points and without consideration to field area". Indeed, the harmonized LUCAS dataset provides information about size of the surveyed parcels in hectares, but this information is limited to four categories (i.e. < 0,5 ha, 0,5 – 1 ha, 1 – 10 ha, > 10 ha), which were not relevant to weight relative importance."

*Comment: 6. Line 158. What's meaning of "relative importance" for LUCAS-based crop sequence? Is it calculated by dividing the total points by the number of points of each crop sequence? Does it include the parcel area information? I think the author need to clarify the calculation methods for "relative importance" through adding mathematical equations. The "relative frequency" also needs to be explained.*

Response: See response to the previous comment

*Comment: 7. Figure 1 shows a step "Extrapolated surfaces from LUCAS data, crop surfaces per county". So how to extrapolate crop surfaces? Are the crop surfaces still point type, or did you upscale to a grid level and further calculated the crop area?*

Response: See response to the previous comment and equation added. For a given (group of) crop, the surface was calculated from total arable area, multiplied by the relative importance of crop sequence types including this (group of) crop and the temporal frequency of this (group of) crops within these crop sequence types.

*Comment: 8. Line 252-257. I agree with the authors' point. These uncertainties resulted from the wrong records from LUCAS datasets.*

*Comment: 9. Line 314-323. The authors discuss about the application of the newly developed crop sequence map in "crop diversity" issues. I suggest the authors can discuss the potential application of crop sequence/rotation map in terrestrial carbon, nitrogen, and hydrological cycles rather than limited in the "crop diversity" field.*

Response: We thank the reviewer for this very constructive comment. We modified the discussion section in lines 313-324 of the revised manuscript to include a broader perspective on the effects of crop rotations. Specifically, we added "soil carbon sequestration" and "water resources management" as crop rotation sensitive issues (lines 313-314), and added in lines

323-324 : "the share of summer and winter crops is relevant to water management as water demand is usually higher during the summer."

*Comment: 10. The authors highlight that the dataset was the first crop sequence map for Europe, but I did not find any information about the spatial resolution. So, I suggest that the authors can upscale the data from point to grid with a reasonable spatial resolution (e.g., 1 km or 0.05 degree), which will be more useful for agricultural studies.*

Response: This is an interesting suggestion, that has been considered. We agree that a gridded map of rotations over the EU would be of interest and would facilitate modeling studies if it could be used as a model input. However, we decided not to develop such a map because gridded agricultural/crop models vary in their spatial resolutions and crops/crop types simulated, which makes difficult to choose a single best spatial resolution and group of crops definition (for example, is soybean part of pulses or oilseeds?). Therefore, at this stage we decided to make publicly available the point-based map of dominant crop sequence types along with the corresponding observed crop (types) in 2012, 2015, and 2018 so that anyone could use these data to do different groupings and describe crop sequence in a way that is relevant to the question addressed.

---

## Author Response (AR3)

Dear Topical Editor,

Thank you for having reviewed our revised manuscript.

Please find bellow a point-by-point reply to your comments.

We hope this reply and the improvements proposed in the revised manuscript will clarify the questions that were raised.

Sincerely yours,

Rémy Ballot, on behalf of the authors
* * *
**Comment:** *In your response letter, you mentioned a code error related to temporary grassland, and you've appropriately adjusted the corresponding figures, tables, and text after resolving the issue. I would like to confirm if this error had any impact on the data product. If it did, please ensure that the data product has been corrected accordingly.*

**Reply:** We confirm that this error impacted the data product. A corrected version was uploaded on July 7. 2023.
* * *
**Comment:** *This manuscript well addresses data quality and potential uncertainties. Could you provide further insights into the implications for future data users, particularly concerning the underestimated importance of grassland crop sequences and the overestimated significance of other crop sequence types?*

**Reply:** We elaborate the implications of this misestimation lines 289-293 of the revised manuscript "As highlighted by previous research (e.g. Martin et al. 2020), crop sequences with temporary grasslands provide specific services such as carbon sequestration, biodiversity conservation or weed control. Future users of our dataset should therefore be aware that this underestimation of the importance of crop sequences with temporary grasslands may result in underestimating such services, as well as overestimating the production from other overestimated crops."